# Molecular signatures of resilience to Alzheimer's disease in neocortical layer 4 neurons

S. Akila Parvathy Dharshini, Jorge Sanz-Ros, Jie Pan, Weijing Tang, Kristen Vallejo, Yu Chen Liu, Marcos Otero-Garcia & Inma Cobos

Selective neuronal vulnerability is a hallmark of Alzheimer's disease (AD), yet the molecular basis of resilience remains poorly understood. Using single-nucleus and spatial transcriptomics to compare neocortical regions affected early (prefrontal cortex, precuneus) or late (primary visual cortex) in AD, we identified a resilient excitatory population in layer 4 of the primary visual cortex expressing *RORB*, *CUX2*, and *EYA4*. Layer 4 neurons in association neocortex shared molecular signatures of resilience. Early-stage resilient neurons upregulated genes associated with synapse maintenance, synaptic plasticity, calcium homeostasis, and neuroprotection (*GRIN2A, RORA, NRXN1, NLGN1, NCAM2, FGF14, NRG3, NEGR1, CSMD1*). We identified *KCNIP4*, which encodes a voltage-gated potassium channel-interacting protein, as a key resilience factor consistently upregulated during early stages of AD pathology. AAV-mediated overexpression of *Kcnip4* in male *App*$^{SAA}$ mice reduced the expression of activity-dependent genes *Arc* and *c-Fos*, suggesting compensatory mechanisms against neuronal hyperexcitability. Our dataset provides a resource for investigating mechanisms underlying resilience to neurodegeneration.

Advancements in single-cell omics have been pivotal in characterizing the transcriptomic diversity of the human neocortex and elucidating selective cell vulnerability in neurodegenerative dementias such as AD[1–6]. Single-nucleus profiling of the neocortex in AD has identified neuronal populations that are vulnerable and depleted early in the disease, such as layer 1 inhibitory interneurons expressing *NDNF/RELN* and layer 2/3 excitatory neurons expressing *CUX2/COL5A2*[2,7,8]. In contrast, few studies have focused on neuronal subtypes that, despite residing in similar microenvironments, remain preserved even in advanced stages of AD. Identifying these resilient subtypes and the mechanisms underlying their preservation could provide valuable insights for therapeutic strategies aimed at slowing disease progression.

We leveraged the progression of AD in the human neocortex— from association cortices to primary cortices[9–12]— to compare early-affected regions (prefrontal cortex, BA9; precuneus, BA7) with late-affected regions (primary visual cortex, BA17) using single-nucleus RNA sequencing (snRNA-seq). Although the neocortex follows a canonical 6-layer pattern, significant quantitative differences exist across regions[13–15]. For instance, layer 4 (L4) is expanded in primary sensory areas, while layers 2/3 and 5 (L2/3, L5) are relatively more prominent in association cortices[3,6,16–19]. Comparing early- and late-affected areas thus provides a robust framework for examining cell-intrinsic and microenvironmental factors influencing selective vulnerability.

Neocortical L4, or the internal granular cell layer, is densely packed with small, granular neurons that serve as major postsynaptic targets of thalamic sensory nuclei and project locally or to nearby cortical regions. Its thickness varies considerably across different cortical areas, comprising 38% of the cortical ribbon in BA17 and 8.6% in BA9. In BA17, also known as the striate cortex, layer 4 contains a distinct band of myelinated fibers called the line of Gennari[17,19]. L4 has

Department of Pathology, Stanford University School of Medicine, Stanford, CA, USA. e-mail: icobos@stanford.edu

long been considered a resilient area in AD due to its lower burden of tau in neurofibrillary tangles (NFTs), although it exhibits amyloid plaques[9,20–22]. However, the composition of L4 at the single-cell level in AD progression remains poorly understood. In an unbiased manner, our study identified a resilient population of L4 neurons in the BA17 characterized by the co-expression of RORB, CUX2, and EYA4. Whether the resilience of these neurons is due to their specific connectivity, molecular properties, or interactions within the microenvironment remains unresolved, underscoring the importance of single-cell approaches in dissecting these complex factors and advancing research into neuronal resilience.

Our dataset comprises snRNA-seq from three neocortical regions (BA9, BA7, BA17) collected from 46 donors representing all stages of disease progression (Braak stages 0–VI). To enrich for neurons, we performed fluorescence-activated nuclear sorting (FANS) for NeuN, resulting in 424,528 nuclei after quality control (QC), of which 362,224 were neuronal. Additionally, we generated single-cell spatial transcriptomics data from 16 tissue sections of BA9 and BA17 obtained from 4 AD and 4 control donors (765,992 cells, after QC). By integrating single-nucleus and spatial transcriptomics, we validated the layer-specific expression of 18 excitatory neuronal subtypes and identified resilient L4 neurons. We employed machine learning methods to validate neuronal subtype annotations across large-scale publicly available AD datasets[5,8,23]. Robust differential gene expression (DGE) analysis, utilizing linear mixed models, bootstrap resampling, and DESeq2 on pseudobulk aggregated counts, identified candidate genes associated with resilience. As proof of principle, we focused on KCNIP4, a gene encoding a voltage-gated potassium channel-interacting protein that regulates neuronal excitability in response to changes in intracellular calcium. We found that KCNIP4 was upregulated in resilient L4 neurons during early disease stages. Furthermore, AAV-mediated delivery of KCNIP4 in a humanized App knock-in AD mouse model (App$^{SAA}$)[24] reduced Arc and c-Fos expression, suggesting potential roles in regulating hyperexcitability. Our dataset is a valuable resource for investigating mechanisms of resilience in neurodegeneration.

## Results

### Neuronal cell type composition during the spatiotemporal progression of AD in the neocortex

In the AD neocortex, neurodegeneration and tau pathology progress from association to primary cortices[9–12]. We profiled nuclear transcriptomes from 243 samples obtained from two association cortices (BA9, BA7) and one primary cortex (BA17) from 46 donors who died at various stages of disease progression and age-matched healthy controls (Braak stages 0–VI). Donor cohorts contributing to each region do not fully overlap, potentially introducing residual confounding in region–pathology associations. Donors were categorized into three pathology groups—low, intermediate, and high (18, 10, and 18 donors, respectively)—based on neuropathological diagnoses using current consensus criteria[10] (Fig. 1a; Supplementary Data 1). For each tissue sample, we collected two single-nucleus suspensions using FANS: one containing all nuclei and one enriched for neurons (NeuN$^+$). In total, we profiled 655,407 nuclei. After rigorous QC to remove nuclei with low gene counts, high mitochondrial content, and doublets, we retained 424,528 high-quality nuclei for downstream analysis (Fig. 1b; Supplementary Fig. 1; Supplementary Data 2). The major cell types included 362,224 neurons (282,930 excitatory and 79,294 inhibitory), astrocytes (14,691), microglia (5071), oligodendrocyte precursor cells (OPCs; 5770), oligodendrocytes (36,589), and vascular cells (183) (Fig. 1c–f).

We identified 18 excitatory (Ex) and 19 inhibitory (In) clusters, corresponding to neocortical neuronal subtypes, using stringent criteria. Our clustering strategy employed unsupervised Leiden clustering, combined with strict thresholds based on silhouette scores and Within Cluster Sum of Squares (WCSS), to enhance clustering reliability and ensure reproducibility. Clusters were named according to canonical markers for major subclasses (CUX2, RORB, THEMIS, and FEZF2 for excitatory; LHX6, ADARB2, PVALB, SST, VIP, and LAMP5 for inhibitory) along with 1–3 top marker genes for each cluster (Fig. 1f, g; Supplementary Data 3). Additionally, we selected gene sets (7–10 genes per subtype) whose combined expression precisely labeled each neuronal subtype across neocortical regions (Supplementary Fig. 2, Supplementary Fig. 3, Supplementary Data 3). The clusters and their marker genes showed consistent expression across BA9, BA7, and BA17. As expected, we observed significant differences in the abundance of neurons in specific excitatory clusters between association cortices and the primary visual cortex, reflecting their different cytoarchitecture[3,13]. For instance, Ex5, characterized by the expression of CUX2, RORB, and EYA4, was overrepresented in BA17 (Fig. 1h). In contrast, all inhibitory clusters were well represented across the three regions (Fig. 1i).

We further assessed cluster reliability by comparing it with that from an AD reference dataset (Seattle Alzheimer's Disease Brain Cell Atlas [SEA-AD]), which includes nearly 1.4 million nuclei from the dorsolateral prefrontal cortex (DLPFC) and uses reference annotations for cell subclasses and supertypes from BICCN (Brain Research through Advancing Innovative Neurotechnologies)[3,23]. We constructed a cosine distance matrix to assess the similarity between the gene expression profiles of both datasets (Fig. 1j). Our annotations closely matched the reference dataset. Additionally, we used the semi-supervised single-cell ANnotation using Variational Inference (scANVI) model to annotate two AD reference datasets (SEA-AD DLPFC[23] and Green and colleagues[5]), based on predictions from our 18 excitatory and 19 inhibitory neuron clusters. Our gene sets consistently labeled the clusters across datasets (Supplementary Figs. 2 and 3).

Glial cell states closely matched those from previous studies[2,5,23,25] and included four astrocyte states, labeled by: SLC1A2/WIF1 (homeostatic), SLC1A2/SMTN, GFAP/CHI3L1/OSMR (reactive) and GFAP/AQP1/VCAN (reactive); four microglia states: CX3CR1 (homeostatic), AIF1 (reactive), CACNA1B (reactive), and CD163 (reactive); and two oligodendrocyte states: OPALIN (myelinating) and COL18A1 (Fig. 1k).

### Spatial distribution of neuronal cell types in association (BA9) vs primary (BA17) cortices

To spatially map the cortical layer distribution of the 18 excitatory and 19 inhibitory clusters in the neocortex of AD and control donors, we performed spatial transcriptomics using the 10x Genomics Xenium platform. We processed four slides containing a total of 16 human brain sections, eight from BA9 and eight from BA17, including samples from four donors with high AD pathology and four age-matched healthy controls (Fig. 2a). All sections comprised the entire neocortical thickness and adjacent white matter. We used the predesigned 266-gene Xenium Human Brain Gene Expression panel, along with a custom 100-gene panel designed to enhance granularity for detecting cortical neuronal subtypes, which included cluster-specific marker genes identified from our snRNA-seq data and public repositories. Additionally, the slides were stained with the 10x Genomics cell segmentation add-on kit to enhance transcript-to-cell assignments.

Our pipeline for cell subtype annotation included two steps. First, we annotated major cell types while simultaneously accounting for transcript signal overlap among closely located cells. To achieve this, we employed four approaches: (1) manual annotation based on k-nearest neighbor graphs, Leiden clustering, and canonical marker genes; (2) heuristic classification with a custom Python script to assign cell types based on the highest expressed transcripts; (3) deep neural network classification via spatialID, trained on the SEA-AD DLPFC dataset; and (4) ingest-based label transfer projecting SEA-AD DLPFC annotations onto the spatial data. We used an ensemble voting strategy to combine predictions from these methods, creating consensus

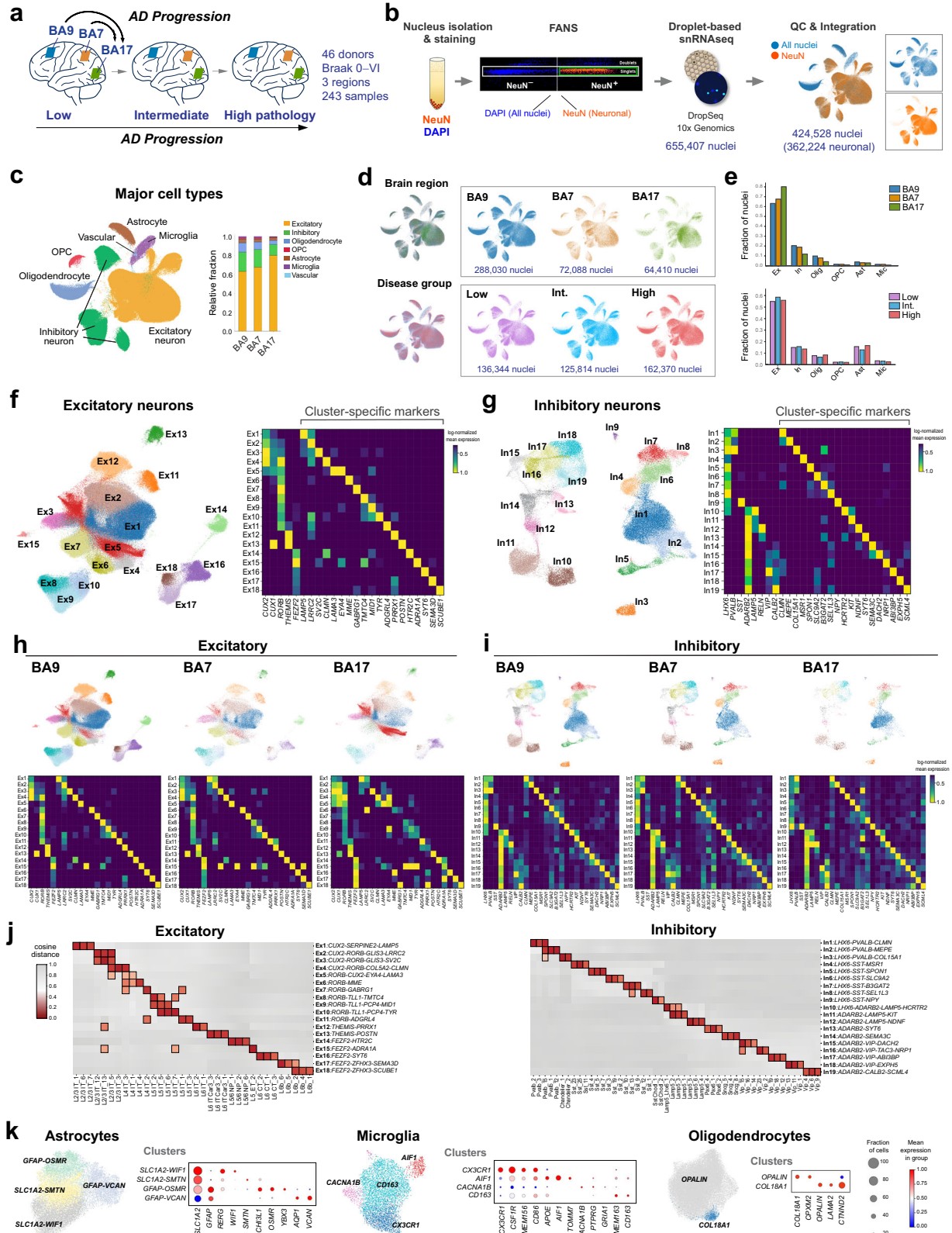

annotations for major cell types and confidence scores. Next, we performed neuronal cell subtype annotation using ingest-based label transfer with our snRNA-seq dataset as a reference.

Our annotated Xenium dataset combining all slides contains 765,992 cells across brain regions and donors (Fig. 2b). Visualization of the 18 excitatory neuronal subtypes in each individual section revealed regional differences between BA9 and BA17, with an overall similar

distribution in AD and controls (Fig. 2c). As expected, there was a higher neuronal density in BA17. The distribution of these subtypes across layers corresponded with their pre-assigned labels and aligned with the layer boundaries indicated by the stains (i.e., DAPI, ribosomal RNA, and αSMA/Vimentin) (Fig. 2d–f). The thickness of L4 varied significantly, comprising over one-third of the cortex in BA17 while accounting for less than 10% of the cortex in BA9, consistent with

**Fig. 1 | Neuronal cell composition across neocortical regions and AD pathology stages. a** Experimental design to study AD progression across neocortical regions and disease stages using snRNA-seq. **b** Neuronal enrichment by FANS, snRNA-seq, and dataset integration yielded 424,528 nuclei (362,224 neurons, after QC). **c**, UMAP and bar plots representing the relative abundance of major cell types. **d** UMAP plots splitting the datasets by region and disease stage group. **e** Fraction of nuclei from each major cell type by region (top) and disease stage group (bottom). **f**, **g** UMAP plots of the annotated excitatory and inhibitory clusters and heatmaps showing the normalized expression of selected subtype and cluster-specific marker genes. **h**, **i** UMAP plots and gene expression heatmaps for each brain region highlighting quantitative differences between association and primary cortices, and overall preserved marker genes across regions. **j** Cosine distance matrix comparing the proximity in gene expression between the excitatory and inhibitory clusters from the SEA-AD DLPFC reference dataset[23] (x-axis) and our BA9 dataset (y-axis). The closer the distance (lower values), the greater the similarity. The top three most closely related clusters are depicted. **k** UMAP and dot plots showing the annotated glial subtypes and states. Source data are provided as a Source Data file.

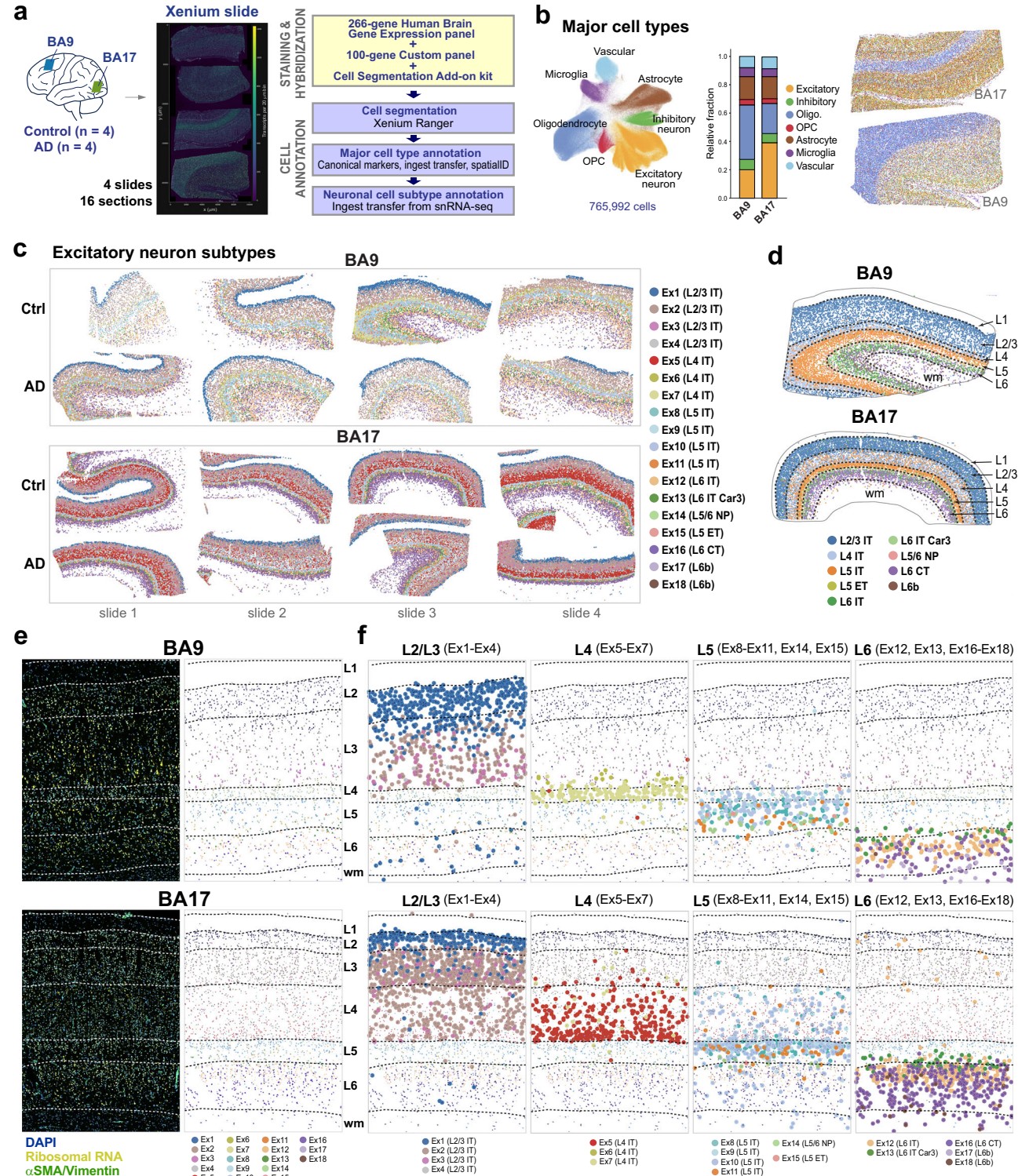

**Fig. 2 | Layer-specific localization of excitatory neuronal subtypes in BA9 and BA17 by Xenium. a** Experimental design for spatial single-cell analysis of neuronal subtypes in fresh-frozen tissue sections from BA9 and BA17 of AD and control donors using Xenium. A representative Xenium slide (slide 2) with four tissue sections (AD-BA17, AD-BA9, Ctrl-BA17, Ctrl-BA9; top to bottom) is shown. **b** UMAP and bar plots depicting the relative abundance of major cell types in the Xenium dataset (765,992 cells, after QC), and representative spatial maps of BA9 and BA17 (slide 2, control donor) after cell segmentation and major cell type annotation. The color coding for major cell types is consistent across all visualizations. **c** Spatial maps of the annotated 18 excitatory clusters across all 16 sections, highlighting differences in neuronal subtype abundance between BA9 and BA17. Small areas corresponding BA18 are excluded. **d**, Representative spatial maps after segmentation and annotation based on reference annotations for excitatory neurons at the cell subclass level, highlighting differences in layer thickness and composition between BA9 and BA17. Dash lines represent boundaries between layers. **e** Representative cortex from control BA9 and BA17 sections showing staining with DAPI, ribosomal RNA (interior RNA staining), and αSMA/Vimentin (interior protein staining) *(left)*, and cell boundaries identified by the multimodal cell segmentation algorithm and annotated using ingest-based label transfer with our snRNA-seq dataset as a reference *(right)*. Clusters are colored according to their identity. Dash lines delineate boundaries between cortical layers. **f** Spatial maps for each excitatory cluster in the areas represented in (**e**), with each cluster overlaying its corresponding cells to highlight their layer distribution and spatial relationships with other excitatory clusters within L2/3, L4, L5, and L6. Source data are provided as a Source Data file.

reference neuroanatomical studies[17]. The composition of L4 also varied significantly, with Ex5 overrepresented in BA17, while Ex6 and Ex7 were overrepresented in BA9, aligning with our snRNA-seq data (Fig. 2f).

These patterns were also observed in an independent spatial dataset generated using 10x Genomics Visium with a different gene panel (Human Neuroscience gene expression panel, with 1186 genes, along with a custom 197-gene panel) (Supplementary Fig. 4). Thus, our integrated single-nucleus and spatial transcriptomics data identified robust clusters characterized by specific marker genes and gene sets, and mapped their spatial laminar distribution across neocortical brain regions.

## Identification of layer 4 excitatory neurons across BA9 and BA17 in AD

Primary cortices, such as BA17, are affected in the latest stage of AD (Braak VI). L4 in BA17 at Braak VI shows amyloid plaques but minimal tau pathology[9,20–22]. Thus, BA17 as a region, and L4 in particular, are considered resilient in AD. However, it remains unclear whether L4 is resilient across neocortical regions. To investigate this, we first identified marker genes for L4 excitatory neuronal subtypes (Ex5, Ex6, and Ex7; L4 IT). Consistent with previous studies, many genes expressed in L4 exhibited spatial gradients extending into layers 3 and 5[3,16,26,27]. L4 was characterized by co-expression of *CUX2* (labeling L2−4) and *RORB* (labeling L3−5), with high expression of *CUX1*. The top cluster-specific markers for Ex5 included *EYA4, KCNH8, LAMA3, VAV3, KCNIP1*, and *TRPC3* (Fig. 3a). While these genes were expressed in BA17, most were detected in only a small subset of cells in BA9. *LAMA3* was expressed in Ex5 neurons across neocortical regions (Fig. 3b). Notably, the Ex5 marker genes were highly conserved in a reference dataset from the mouse neocortex[28] (Fig. 3c). Ex6 and Ex7 exhibited high expression of *RORB* and low expression *CUX2*, with Ex6 expressing *MME* and Ex7 expressing *GABRG1* (Fig. 3a, b). Double fluorescent RNAscope in situ hybridization (ISH) for *EYA4, MME*, or *GABRG1*, along with *SLC17A7*, in BA9 and BA17 control tissue sections confirmed their expression in L4 (Supplementary Fig. 5). The expression of a subset of L4 markers (*CUX1, RORB, CUX2, EYA4, KCNH8, TRPC3*, and *VAV3*) in Xenium sections confirmed their relative specificity for labeling Ex5, Ex6, and Ex7 populations (Fig. 3d).

Visualization of Ex5, Ex6, and Ex7 in the Xenium sections highlighted the spatial distributions of each cluster in L4 of BA17 and BA9 (Fig. 3e). In BA17, Ex5 exhibited a gradient in cell density, with higher density deeper in the layer and a sharp boundary with L5, while Ex5 cells mixed with L2/3 cells superficially. Ex5 cells were underrepresented in BA9, and their abundance varied considerably across samples in both controls and AD cases. In BA9, Ex6 cells were located at the boundary between L4 and L3, positioned deeper than the more abundant Ex7 cells. Ex6 cells were rare in BA17. Notably, five sections from BA17 also contained adjacent BA18, an association-type cortex. In BA18, Ex5 cells were more abundant, while Ex6 cells were less abundant compared to BA9 (Fig. 3f). Although the relatively low number of cases

limits robust comparisons across regions in BA9, BA17, and BA18 in AD and control samples, these observations suggest variations in cell composition in L4 that may reflect functional specializations across regions.

ISH for *EYA4* and *KCNH8* in human BA17 tissue sections from the Allen Human Brain Atlas[29] also confirmed their expression in L4 granule neurons. The highest expression was observed in deep layer 4c (Fig. 3g, h), while expression in layers 4a and 4b was low. Notably, the most commonly used laminar nomenclature distinguishes three sub-layers within L4: 4a, 4b, and 4c, although some authors classify layers 4b and 4c as part of L3, a view supported by tract-tracing studies in macaques[30,31]. The expression of VGLUT2, which labels presynaptic terminals from the lateral geniculate nucleus (LGN) projecting to L4 in BA17 across species[32,33], matched the expression of *EYA4* and *KCNH8* (Fig. 3h). Thus, *EYA4* and *KCNH8* preferentially label what is considered layer 4 proper in BA17.

To identify our L4 clusters in external datasets, we used scANVI to predict our annotations in three reference datasets from the prefrontal cortex (SEA-AD DLPFC[23]; Green et al., 2024[5]; Mathys et al., 2023[8]) and one from the primary visual cortex (Jorstad *et al.*, 2023[3]) (Supplementary Fig. 6). We observed high similarity across datasets originating from the same brain region based on cosine distance scores, the expression of cluster-specific markers, and by plotting author-annotated and predicted clusters (Supplementary Fig. 6c–g). As expected, the number of Ex5 cells predicted in the BA17 reference dataset was high: 63,870 cells (34.42%) out of 185,565 excitatory cells. In contrast, it was low in the prefrontal cortex reference datasets: 2152 cells (0.33%) out of 660,751 excitatory cells in the SEA-AD dataset; 19,360 cells (3.03%) out of 637,968 in Green et al.; 3,361 cells (3%) out of 112,143 in Mathys et al., compared with 7943 cells (4.36%) out of 182,140 in our BA9 dataset. Ex5 cells were most closely related to supertypes L4 IT_2, L4 IT_3, L4 IT_5, and L4 IT_6 from Jorstad et al., 2023[3] (WithinArea_clusters) (Fig. 3l). In contrast, Ex6 (SEA-AD supertype L4 IT_4) and Ex7 (SEA-AD supertype L4 IT_2) were well represented in the prefrontal cortex across datasets and underrepresented in the BA17 dataset (Fig. 3h, k). Thus, comparisons across independent datasets showed consistent alignment of our L4 excitatory neuron annotations. Together with Xenium data showing Ex5 enrichment in BA17 and rarity in BA9, these cross-dataset mappings support defining Ex5 as a BA17-enriched L4 IT population specialized for the primary visual cortex, with a shared molecular signature and variable prevalence across neocortical regions.

## Relative preservation of layer 4 excitatory neurons during AD progression

To investigate the vulnerability of L4 excitatory neurons to AD progression, we used scCODA[34] and a generalized linear mixed model (GLMM) to model neuronal composition across low, intermediate, and high pathology groups in BA9 and BA17. We controlled for covariates such as sex, age, *APOE* genotype, and profiling assay (Fig. 4, Supplementary Data 4–6). The scCODA analysis revealed a significant relative

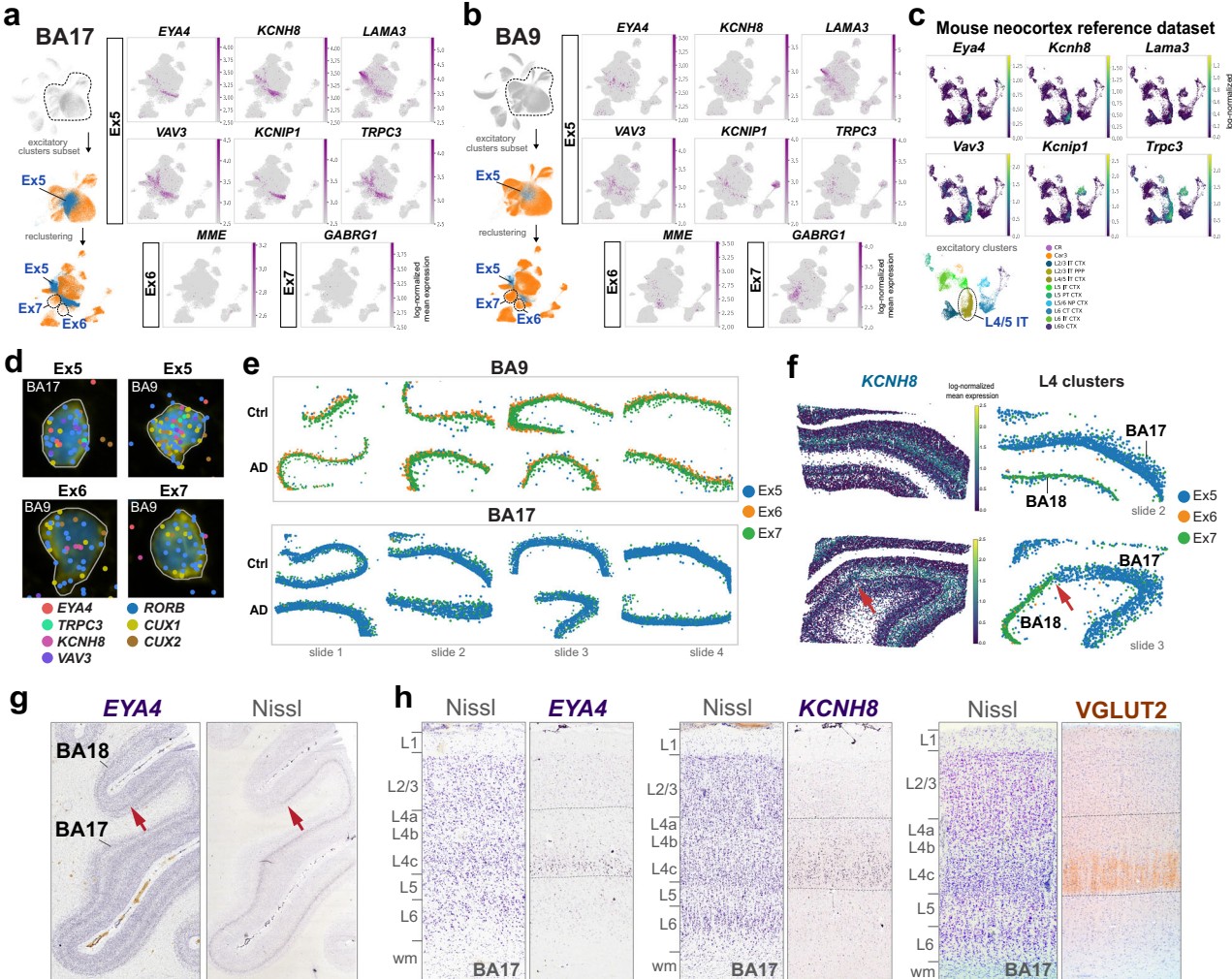

**Fig. 3 | Markers of layer 4 across neocortical regions.** UMAP plots highlighting the top L4 marker genes in BA17 (**a**) compared to BA9 (**b**). The Ex5 cluster (blue) and its top marker genes (*EYA4, KCNH8, LAMA3, VAV3, KCNIP1, TRPC3*) are over-represented in BA17, whereas Ex6 (*MME*) and Ex7 (*GABRG1*) are overrepresented in BA9. **c** UMAP plots from mouse neocortex snRNA-seq[28] highlighting the conserved expression of top Ex5 marker genes in a cluster annotated as L4/5 intratelencephalic (IT). **d** Representative L4 cells and their top marker genes in Xenium. Transcripts (colored dots) are overlaid on their corresponding cells (stained with DAPI and ribosomal RNA), with the cell boundaries delineated (gray lines) by the Xenium cell segmentation algorithm. **e** Spatial maps of the annotated L4 excitatory clusters across Xenium sections, highlighting the relative abundance of Ex5 (blue) in BA17 and of Ex6 (orange) and Ex7 (green) in BA9. BA18 areas are excluded. **f** *KCNH8* expression map (left) and spatial maps of L4 clusters (right) in representative

occipital cortex Xenium sections containing BA17 and adjacent BA18 (primary and secondary visual cortex, respectively; red arrow indicates the transition between BA17 and BA18) highlighting differences between primary and secondary cortices. Identification of Ex5 neurons in L4 of BA17 histological sections. Low-magnification images of the occipital cortex at the transition between BA17 and BA18 (the red arrow in (**g**) indicates the transition between BA17 and BA18) highlight the abundance of *EYA4*+ cells in BA17 (Allen Human Brain Atlas, https://human.brain-map.org/ish/experiment/show/80510718). Higher magnification images of BA17 (**h**) show the expression of *EYA4* and *KCNH8* in L4 (Allen Human Brain Atlas, https://human.brain-map.org/ish/experiment/show/78937929). The boundaries of L4 are defined histologically in parallel Nissl-stained sections and by the expression of VGLUT2 in the terminals of thalamocortical projections from the LGN.

increase in the proportion of Ex5 neurons in high compared to low pathology cases in both BA9 (log2-fold change = 1.75) and BA17 (log2-fold change = 0.46) (Fig. 4a). This suggests the Ex5 population is resilient and becomes more prominent as other neuronal subtypes are lost. The GLMM, which modeled proportional abundance using a beta distribution, supported this finding, showing a significant increase in Ex5 neurons in BA9 (FDR = 0.008) and a similar, non-significant trend in BA17 (Fig. 4b). Because BA17 samples were predominantly sequenced using Drop-seq, the observed compositional shifts in this region may reflect platform-specific biases, despite cross-platform integration and covariate adjustment.

To address potential technical biases, we performed two additional analyses. First, to confirm that our findings were not an artifact of lower transcript counts or shifts in gene expression among L4

clusters, we conducted the same analyses on a filtered dataset with a minimum of 500 genes per cell using reference annotations at the cell subclass level. The total L4 IT population remained relatively increased in high-pathology cases in both BA9 (log$_2$-fold change = 0.21) and BA17 (log$_2$-fold change = 0.33) (Supplementary Fig. 7). Second, since Ex5 neurons have smaller cell bodies and lower gene counts, we evaluated whether our initial QC filter (<300 genes) excluded a significant portion of them. We selected previously filtered neuronal nuclei with gene counts ranging from 200 to 300 (62,498 nuclei) and used scANVI to predict their identity, using our dataset as a reference. After incorporating 48,849 nuclei (78%) that were confidently assigned (99% probability) to annotated clusters, we found that the overall neuronal composition remained unchanged, and the relative preservation of Ex5 neurons remained statistically significant (Supplementary Fig. 8).

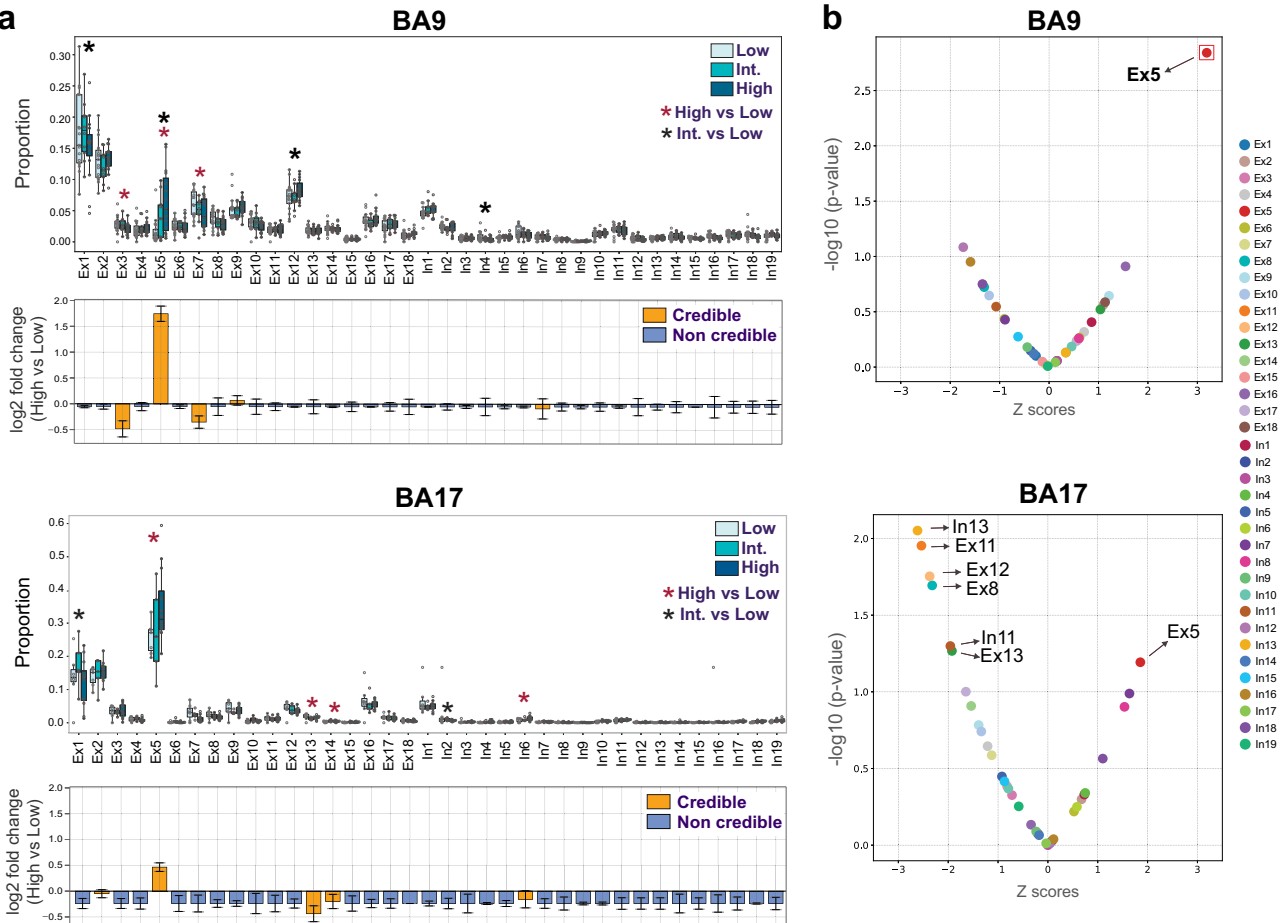

**Fig. 4 | Relative preservation of Ex5 neurons in advanced AD. a** Boxplots showing neuronal cell composition estimated with scCODA across pathology disease groups in BA9 and BA17. Individual donor proportions are overlaid as open circles. Data are presented as median (center line) and interquartile range (IQR; box limits); whiskers extend to the most extreme values within 1.5×IQR. Circles beyond the whiskers represent outliers. Sample sizes for BA9: low 17, intermediate 10, high 15 donors; BA17: low 7, intermediate 5, high 12 donors). Credible differences between high and low pathology groups (red asterisks) and between intermediate and low groups (black asterisks) are shown for clusters with a magnitude of change (log2-fold change) greater than 0.1, in either direction. Credible effects were defined at those with a posterior inclusion probability (PIP) > 0.95. The lower plots show the credible effects (highlighted in orange) along with the fold changes between high and low pathology groups; bars represent log2-fold change, and error bars indicate the standard error of the mean. **b** Differential cell proportion analysis of neuronal populations between low and high disease groups using GLMM in BA9 and BA17. Ex5 neurons showed increased relative abundance in advanced AD in BA9 (FDR = 0.008). In BA17, Ex5 neurons showed a non-significant trend of increase (*p*-value = 0.06), while reductions were observed in deeper-layer excitatory populations, including Ex8 (L5 IT; *p*-value = 0.02), Ex11 (L5 IT; *p*-value = 0.01), Ex12 (L6 IT; *p*-value = 0.01), and Ex13 (L6 IT Car3; *p*-value = 0.05), though the changes did not reach statistical significance after FDR correction. Source data are provided as a Source Data file.

Our analyses also identified vulnerability in other neuronal populations, including Ex3 neurons in BA9 (large deep L3 neurons expressing *SV2C*) (Fig. 4a), L5 IT in BA9 (Supplementary Fig. 7a), and specific interneuron clusters expressing SST (In4 in BA9; In6 in BA17; Fig. 4a). Although these changes were less robust and consistent across the analyses (scCODA and GLMM) and annotation methods, they aligned with reported findings of L2/3 IT and SST-expressing interneurons vulnerability from high-quality association neocortex datasets[2,4,7,8,23,35].

In summary, our data consistently show that the L4 IT excitatory neuron population is relatively preserved during AD progression in BA9 and BA17. Within this population, the Ex5 subtype is particularly resilient, becoming increasingly prominent as neighboring neurons degenerate.

### Differential gene and pathway expression in vulnerable vs resilient neocortex in AD

To identify genes and pathways altered during disease progression in vulnerable and resilient regions, we performed DGE analysis comparing two disease stages ('early': low vs. intermediate pathology; 'late': intermediate vs. high pathology) and two neocortical regions (BA9 and BA17) for each neuronal subtype. Given AD progression, we expect that gene expression changes observed in late-stage BA17 will be concordant with those seen in early-stage BA9. Statistical power to detect differentially expressed (DE) genes is influenced by technical and biological factors, such as the number of nuclei, sequencing depth, RNA integrity, and age-dependent epigenetic changes[36,37]. To address the heterogeneity of the samples and ensure the reliability of our findings, we applied several DGE methods, including a linear mixed model implemented in MAST and lme4, bootstrap resampling with 100 iterations, and DESeq2 on pseudobulk aggregated counts (Fig. 5a, Supplementary Fig. 9). We defined 'high-confidence' DE genes as those consistently identified across methods.

The total number of DE genes was higher in BA9 compared to BA17 and in the 'late' disease groups compared to the 'early' groups, reflecting gene expression changes associated with AD progression (Fig. 5b,c, Supplementary Data 7). Subtypes previously identified as vulnerable, such as L2/3 IT excitatory neurons (Ex1, Ex2), exhibited

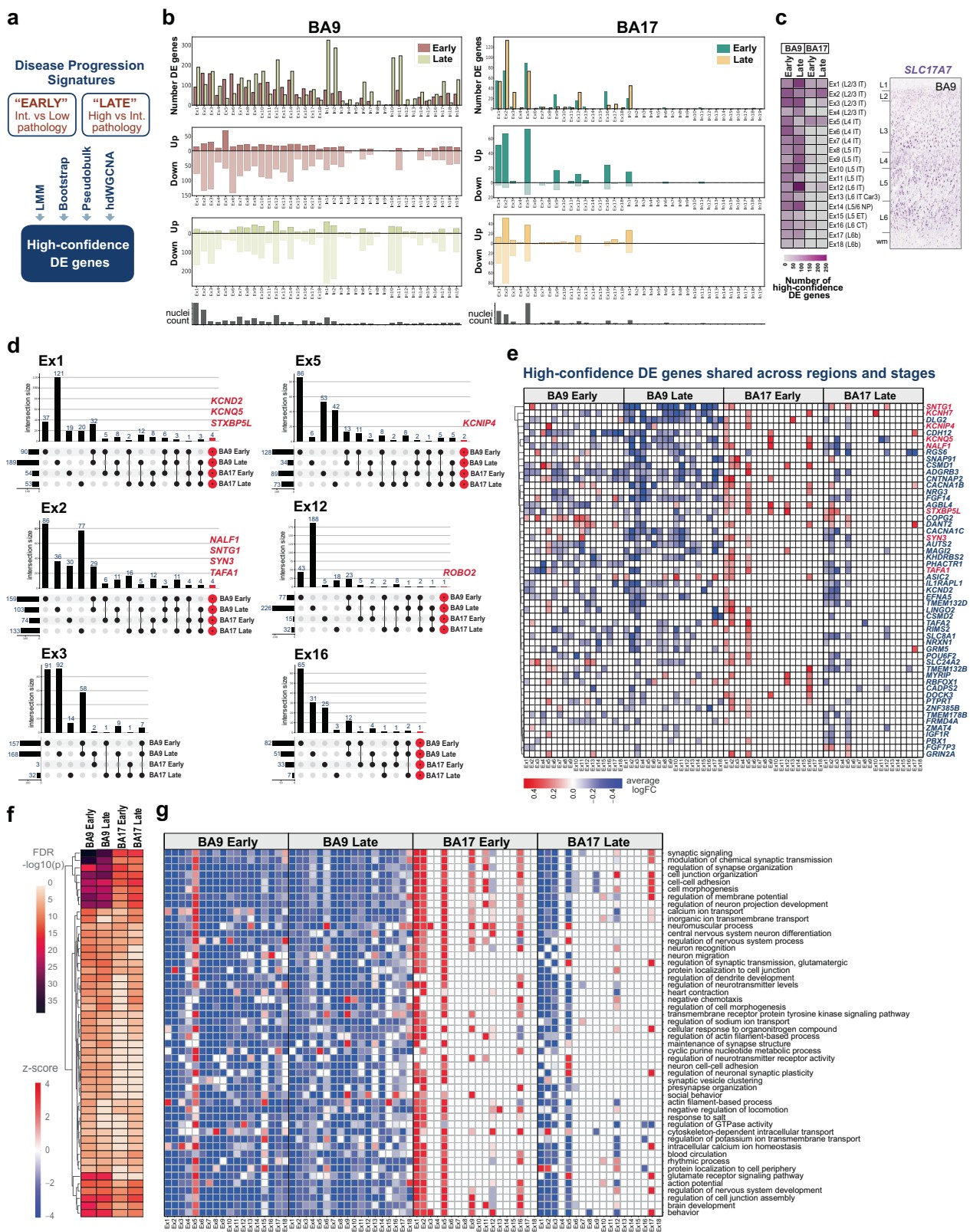

more DE genes across both regions and disease stages. However, in BA9, some excitatory clusters, including the vulnerable Ex2 (L2/3 IT) and the resilient Ex5 (L4 IT), showed more significant changes in the 'early' compared to 'late' stages. Most DE genes in BA9 were downregulated, except for Ex5, where over 50% were upregulated in the 'early' stages. In contrast, in BA17, the majority of DE genes were

upregulated, especially in the 'early' stages, in both vulnerable (Ex2) and resilient (Ex5) subtypes (Fig. 5b).

A total of 986 genes were categorized as 'high-confidence' DE genes. To distinguish between genes that were shared or unique across brain regions and disease stages (i.e., BA9-Early, BA9-Late, BA17-Early, BA17-Late), we generated UpSet plots showing intersections among

**Fig. 5 | Transcriptome signatures of AD progression in neocortex. a** 'High-confidence' DE genes were identified using a linear mixed model and either bootstrap, pseudobulk, or hdWGCNA. 'Early' and 'late' DE genes correspond to intermediate vs. low and high vs. intermediate AD pathology, respectively. **b** Bar plots show total numbers of DE genes, upregulated genes, and downregulated genes, identified by a linear mixed model. Downregulation predominates, though early-stage BA17 shows high upregulation. Nuclei counts per cluster are provided. **c** Heatmap of high-confidence DE genes in BA9 and BA17 excitatory clusters. DE gene counts increase with pathology progression and from BA9 to BA17. *SLC17A7* ISH staining shows layer distribution for reference. **d** UpSet plots show intersecting high-confidence DE genes across regions and stages for six excitatory neuronal subtypes. Rows correspond to each of the four conditions, and columns represent the intersections. Genes highlighted in red are differentially expressed in all four conditions. **e** Heatmap of 54 high-confidence DE genes shared across brain regions and disease stages in excitatory neuronal subtypes. Only DE genes shared in at least

5 clusters are represented. Colors indicate the average log-fold change obtained from the linear mixed model. Hierarchical heatmap visualization of functional enrichment analysis (**f**) in excitatory neurons from BA9 and BA17 at early and late stages highlights the common biological pathways enriched across regions and disease stages. High-confidence DE genes were used as input for gene ontology. The top 50 enriched pathways are represented. Heatmap visualization of the enriched pathways within each excitatory neuronal subtype (**g**) shows gene downregulation in most subtypes from BA9 at both early and late stages and in BA17 L2-3 excitatory IT neurons (Ex1-3) at late stages, and gene upregulation at early stages in BA17. Ex5 from both BA9 and BA17 at early stages shares enriched pathways with upregulation in gene expression. Pathway-level values represent the net directional bias among term-associated high-confidence DE genes within each comparison and do not imply uniform regulation of all genes within a pathway. The z-score values represent changes in gene expression. Source data are provided as a Source Data file.

these four conditions for each excitatory neuronal type (Fig. 5d). Although most genes were unique, likely due to the stringent criteria used to define 'high-confidence' DE genes, 15–27% were shared across at least two conditions within clusters with a high number of nuclei (Ex1, Ex2, Ex5, Ex12). The overlap of DE genes was greater within a single region across disease stages than it was across different brain regions. This supports that vulnerability and resilience factors are influenced by both region-specific cell identity and the local microenvironment. Nonetheless, in the Ex5 cluster, 19 DE genes were common between BA9 and BA17 at early disease stages, and nine were common at late disease stages.

We identified 54 high-confidence DE genes common across all four conditions. Heatmaps of their expression changes revealed a consistent pattern: greater changes in BA9 compared to BA17, with downregulation increasing with disease progression in BA9 and upregulation shifting to downregulation with disease progression in BA17 (Fig. 5e). Genes exhibiting this pattern included *KCNH7, KCNQ5, DLG2, SNTG1, NALF1, CNTNAP2, FGF14, AUTS2,* and *MAGI2*. In contrast, a few genes, such as *COPG2* and *SLC24A2*, were upregulated at early stages in both BA17 and BA9. Notably, several high-confidence DE genes have previously been identified as genetic risk factors for AD, including *CSMD1, NRG3, SYN3, NRXN1, SLC24A2, DLG2,* and *KCNIP4*[38–43].

In a similar DGE analysis using BINCC reference annotations for excitatory subclasses on a filtered dataset with a minimum of 500 genes per cell, we identified a total of 962 'high-confidence' DE genes, with 460 overlapping between both approaches. Of these 962 genes, 35 were shared across all four conditions, including *CSMD1, NRG3, SLC24A2, DLG2,* and *KCNIP4* (Supplementary Fig. 10, Supplementary Data 7).

Pathway enrichment analysis revealed shared pathways across regions and stages, including those involved in regulating synaptic organization, membrane potential, neurotransmitter levels, ion (calcium, sodium, and potassium) transport, intracellular calcium homeostasis, glutamate receptor signaling, synaptic vesicle clustering, and cell-cell adhesion (Fig. 5f,g; Supplementary Data 8). The same pattern persisted: enrichments were more significant in BA9 compared to BA17, and genes within the involved pathways were generally downregulated, except in the resilient regions (BA17-Early) and resilient neuronal subtypes (Ex5) (Fig. 5g).

### Genes and pathways associated with resilience in the AD neocortex

To further define genes and pathways associated with resilience, we compared two neuronal subtypes: prototype vulnerable neurons (Ex2; L2/3 IT) and resilient neurons (Ex5; L4 IT) (Fig. 6a,b, Supplementary Data 9). We hypothesized that resilience-associated genes would be enriched and upregulated in Ex5, particularly at early stages and in BA17, consistent with disease progression and the preservation of L4 in

AD. 'High-confidence' genes upregulated in Ex5 neurons at early stages in both BA9 and BA17 inlcuded: *CSMD1*, which encodes a synaptic protein that protects against complement-mediated synapse elimination[44]; *GRIN2A, GRM7, PTPRT,* and *KCNIP4*, which are involved in regulating neuronal excitability, synaptic transmission, synaptic organization, and synaptic plasticity; *SLC24A2*, a member of the calcium/cation antiporter superfamily involved in calcium homeostasis; *UBE2E2*, encoding an E2 ubiquitin-conjugating enzyme; *LINGO2*, a negative regulator of neuronal growth and survival; *TAFA1* and *TAFA2*, homologous genes encoding chemokine-like proteins with roles in neuronal survival; and *AUTS2*, involved in transcriptional activation and actin cytoskeleton reorganization. Some of these genes, such as *CSMD1, GRIN2A,* and *PTPRT*, were also upregulated in Ex2 and other excitatory neuronal subtypes at early stages in BA17, suggesting shared neuroprotective roles across different neuronal subtypes. Other DE genes upregulated early in BA17 and involved in synapse organization and function included: *CSMD2, NRXN1, NRG1, NRG3, TENM2, CACNA1B, GRID2, SLC8A1, SYN3, DLG2, DLGAP1, STXBP5L, NCAM2,* RIMS2, and *ADGRB3*. Additionally, genes upregulated at early stages in Ex5 included those encoding neurotrophic factors and proteins with neuroprotective properties, such as *NRG3, FGF14,* and *NCAM2* (Fig. 6a, b, Supplementary Data 9).

Next, we analyzed high-dimensional weighted gene co-expression network analysis (hdWGCNA) data to compare systems-level changes in vulnerable (Ex2; L2/3 IT) and resilient neurons (Ex5; L4 IT) (Fig. 6c, d, Supplementary Data 10). In Ex5 neurons from BA17, we identified two candidate resilient modules, M2 and M3, where network genes were predominantly upregulated at early disease stages. The top 10 hub genes in these modules are: *KCNIP4, CADM2, NRG3, ADGRB3, NRXN1, NALF1, NEGR1, FGF14, TENM2,* and *CUX1* (for M2), and *PTPRD, LRRC4C, CNTN5, RORA, ANKS1B, NLGN1, RALYL, IQCJ–SCHIP1, SNTG1,* and *RIMS2* (for M3). For Ex5 neurons from BA9, we identified three candidate resilient modules: M2, M3, and M4 (Fig. 6c). A biological function network representation of these hdWGCNA genes, integrating the candidate resilience modules BA17–M2, M3 and BA9–M2, M3, M4, underscored the potential roles of trans-synaptic signaling, calcium homeostasis, and neuronal excitability in resilience. Relevant genes within these modules include *GRIN2A, GRM5, GRM7, CACNA1B, CACNA1C, CACNG5, KCNIP4, NALF1, NRXN1, NLGN1, NRG3, PTPRD,* and *FGF14* (Fig. 6d).

### Increased *KCNIP4* expression is associated with resilience in AD
We focused on *KCNIP4*, a gene specifically upregulated in resilient Ex5 neurons at early disease stages in both BA17 and BA9 (Fig. 6a), as a proof of principle to validate our approach for identifying genes associated with resilience. This gene encodes a voltage-gated potassium channel-interacting protein (KCHIP4 or KCNIP4) that regulates neuronal excitability. KCNIP4 also interacts with Presenilins and has

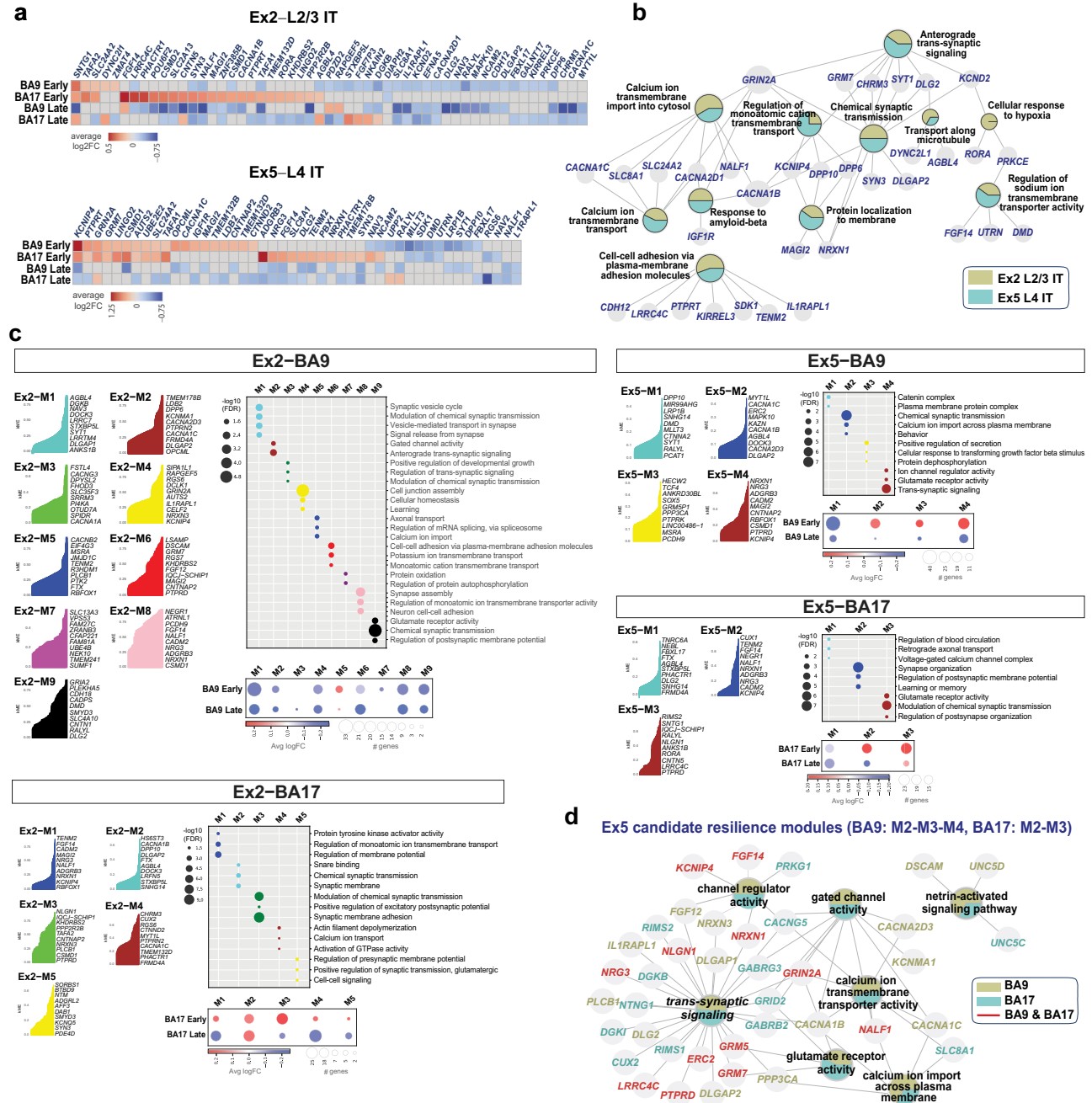

**Fig. 6 | Transcriptome signatures of resilience in Ex5 L4 IT neurons. a** Heatmaps displaying 'high-confidence' DE genes shared across BA9 and BA17 at early and late stages in prototype vulnerable excitatory (Ex2; L2/3 IT) and prototype resilient (Ex5; L4 IT) neuronal subtypes. Genes differentially expressed in at least two of the four comparisons are depicted. Heatmaps are colored based on log2 fold change values. **b** Biological function network of the genes represented in (**a**). Colored nodes represent gene sets of biological functions contributed by the vulnerable (Ex2) and resilient (Ex5) subtypes. Node size reflects the number of connections between biological functions (minimum number = 5). **c**, Co-expression networks for vulnerable (Ex2; L2/3 IT) and resilient (Ex5; L4 IT) neuronal subtypes from BA9 and BA17, identified by hdWGCNA. The top 10 intra-module connected genes, ranked by Kme, for each module are represented. The enrichment dot plot illustrates the top functional categories of genes within each module. The color of the dots indicates the module, while the size of the dot reflects the significance of the enrichment. The gene expression dot plots represent the average logFC for each module at 'early' and 'late' disease stages. The size of the dot represents the number of differentially expressed genes, and the color indicates the magnitude of expression changes. **d** Enrichment network for candidate resilient modules in Ex5 L4 IT neurons. The top 50 highly co-expressed genes from modules M2, M3, and M4 (BA9) and modules M2 and M3 (BA17), along with their enriched biological functions, are shown. Colors represent contributions from BA9 (moss), BA17 (teal), or both (red), along with their enriched biological functions.

been previously linked to AD[45,46]. Our analysis showed that *KCNIP4* is predominantly expressed in excitatory neurons (except Ex14; L5/6 NP) and OPCs (Fig. 7a), as well as a microglia cluster characterized by high expression of synapse-related genes (cluster Microglia-Reactive-*CAC-NA1B;* Supplementary Data 3). Using a linear mixed model (implemented using the MAST package)[47], we estimated *KCNIP4* expression

across disease stage groups. After controlling for fixed covariates (assay, sex, RIN, and total counts) and random effects (donor), we consistently observed increased *KCNIP4* expression in Ex5 neurons as disease progressed (Fig. 7b).

To quantify KCNIP4 protein levels in resilient versus vulnerable neurons, we performed immunohistochemistry for KCNIP4, EYA4, and

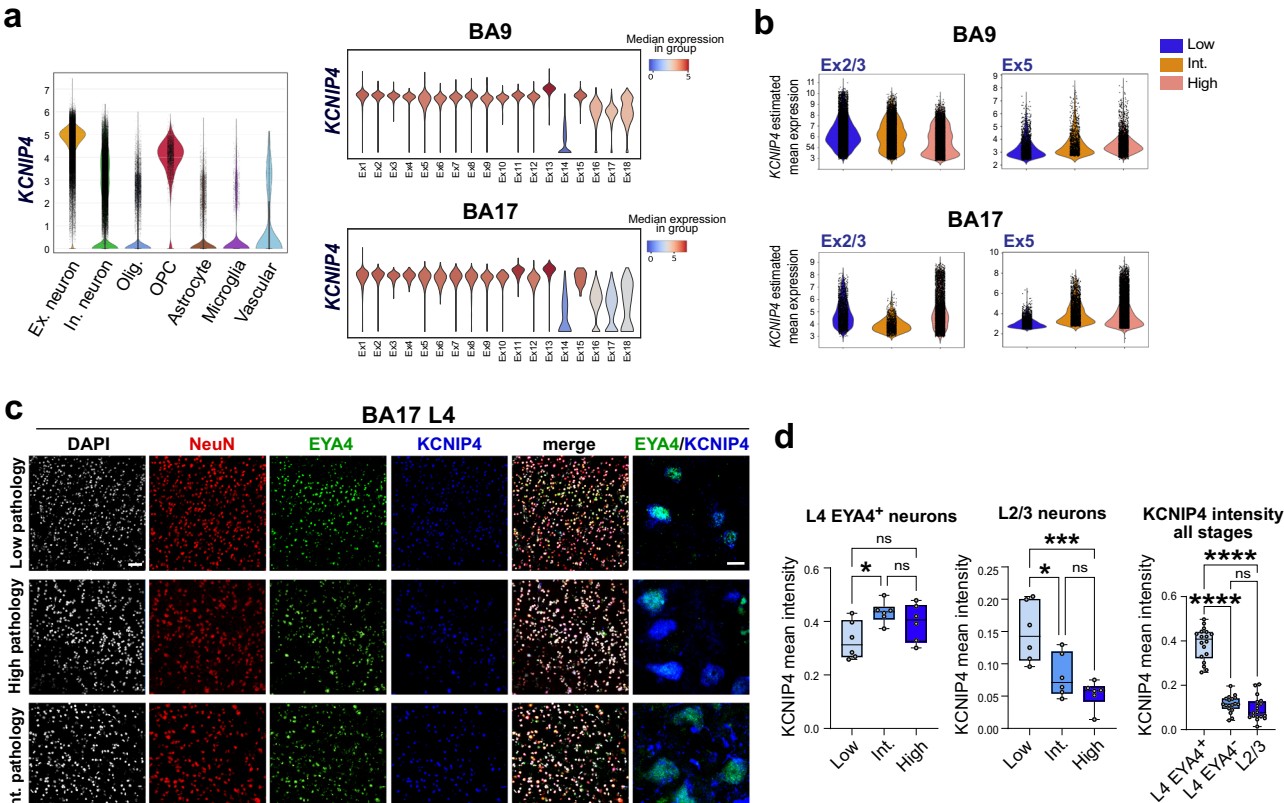

**Fig. 7 | KCNIP4 upregulation in resilient L4 neurons. a** Violin plots showing *KCNIP4* gene expression across major cell types (left) and excitatory neuronal subtypes from BA9 and BA17 (right). **b** Violin plots showing *KCNIP4* expression across AD disease groups in Ex2 and Ex5 neurons from BA9 and BA17. Log-normalized expression levels of KCNIP4 are shown. **c** Immunostaining for KCNIP4, EYA4, and NeuN in cryosections from low, intermediate, and high pathology stages illustrating increased expression of KCNIP4 in L4 EYA4+ neurons in BA17. **d** Quantification of KCNIP4 protein expression levels in L4 EYA4+ neurons, L4 EYA4−

neurons, and L2/3 neurons from BA17 across disease stages ($n = 6$ donors per disease group). Data are shown as median ± IQR; whiskers represent minimum and maximum values. One-way ANOVA with two-sided Tukey's test was used for multiple comparisons (*$p$-value < 0.05; ***$p$-value < 0.001; ****$p$-value < 0.0001; exact $p$-values are available in the Source Data file). Scale bars: 200 μm for low magnification images; 30 μm for high magnification images. Source data are provided as a Source Data file.

NeuN in sections of BA17 from low, intermediate, and high pathology groups (Fig. 7c). EYA4 labels L4 granule cells in the cerebral cortex and is also expressed by a subset of GABAergic interneurons, which are sparse and located predominantly in the superficial layers. The mean intensity of KCNIP4 in neuronal somas was significantly higher in L4 EYA4+ neurons at intermediate disease stages compared to controls, and lower in supragranular (L2/3) neurons at intermediate and high stages compared to controls (Fig. 7d).

KCNIP4 is an integral component of Kv4 channel complexes and belongs to the EF-hand family of small calcium-binding proteins. Like other Kv channel-interacting proteins, it may control neuronal excitability by regulating A-type outward potassium currents[48]. Thus, we hypothesized that increased KCNIP4 expression may reduce neuronal hyperexcitability in AD. To investigate this, we used AAV to overexpress Kcnip4 in excitatory neurons. We generated the AAV vector PHP.eB-CaMKIIa-Kcnip4-P2A-EGFP, using the PHP.eB serotype to efficiently transduce neurons in the CNS, the CaMKIIa promoter to selectively target excitatory neurons, the mouse *Kcnip4* transcript, and EGFP as a reporter. As a control, we used the same AAV containing only EGFP (Fig. 8a). First, we overexpressed *Kcnip4* in primary mouse cortical neurons prepared from postnatal day 0 (P0) pups and assessed neuronal activity using calcium imaging. Neurons were co-transduced with either *Kcnip4* AAV or control GFP AAV, along with PHP.eB-Syn.NES-jRGECO1a.WPRE.SV40 to enable real-time calcium imaging. At DIV12, neurons were treated with 200 nM amyloid-β 1–42

(Aβ1–42) oligomers to increase intracellular calcium levels, or vehicle as a control, for 48 h (Fig. 8a). Calcium imaging at DIV14 revealed that neurons transduced with *Kcnip4* exhibited a significant reduction in spontaneous activity, as evidenced by decreased Ca²⁺ transient events frequency, both under basal conditions and following Aβ1–42 oligomers treatment, compared to control neurons expressing GFP alone (Fig. 8b, c). To confirm that the observed effects on neuronal activity were not due to AAV-related toxicity, we performed a TUNEL assay on the in vitro preparations and found no TUNEL+ neurons in either the GFP or Kcnip4 transduced neurons (Supplementary Fig. 11a). These findings suggest that Kcnip4 overexpression attenuates neuronal hyperactivity, even in the presence of elevated Aβ1–42 oligomers.

We then evaluated *Kcnip4* overexpression in vivo using a humanized *App* knock-in mouse model of familial AD (*App*^SAA KI/KI)[24] (Fig. 8d). To assess the ability of the *Kcnip4* AAV to increase KCNIP4 protein levels in the mouse brain, we performed Western blotting on cortex tissue lysates from 12-month-old WT mice treated with 2 different doses of *Kcnip4* AAV (5 × 10^10 vg and 1 × 10^11 vg, retro-orbitally). Mice treated with the higher dose showed a significant increase in KCNIP4 (Fig. 8e). We injected 12-month-old homozygous *App*^SAA, which exhibit amyloid plaques, microgliosis, and plaque-associated dystrophic neurites[24], with either *Kcnip4* AAV or control AAV (1 × 10^11 vg, retroorbitally). WT mice from the same genetic background and age also received both AAVs. Mice were sacrificed,

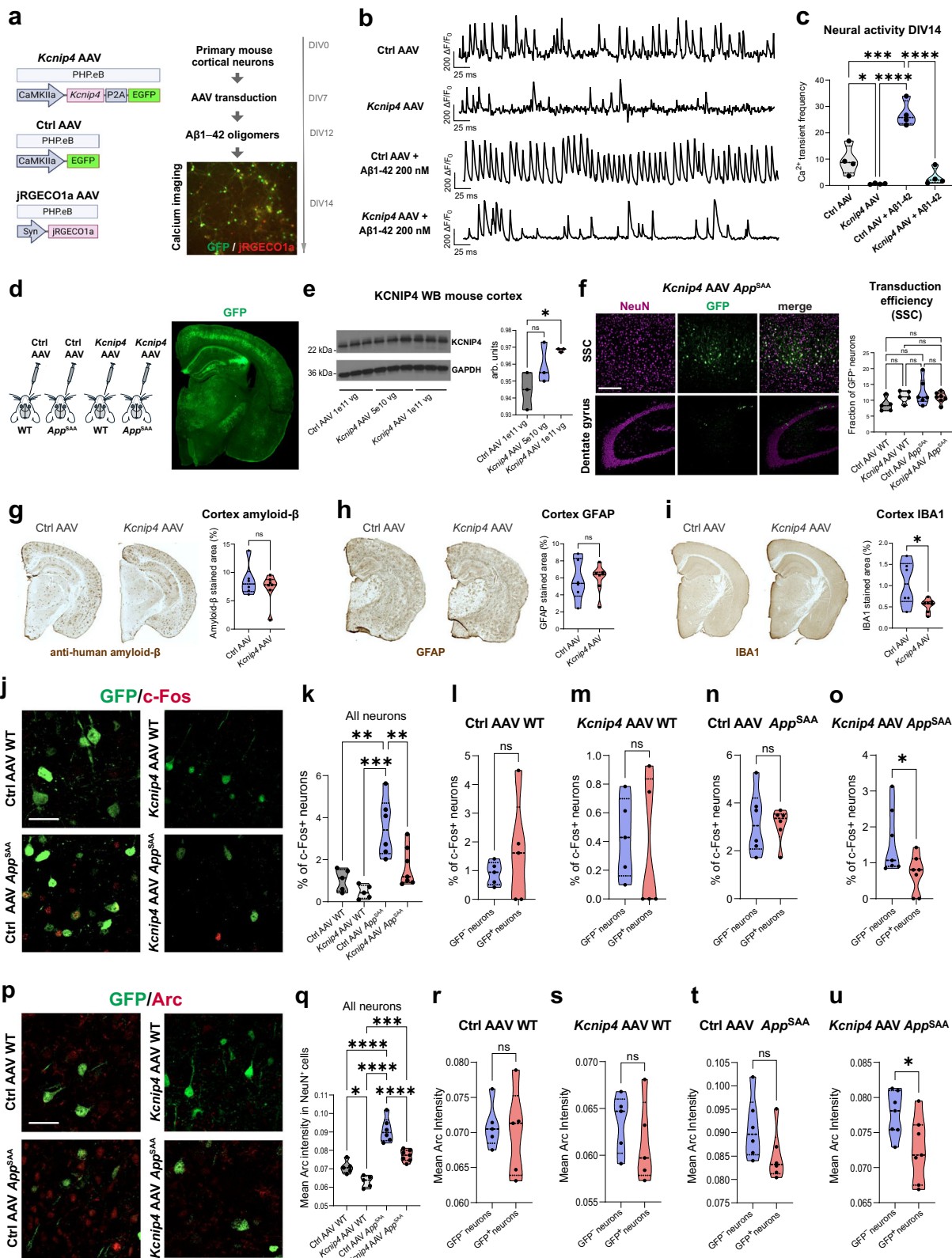

and brain tissue was collected one month after injection. GFP+ neurons were detected throughout the cerebral cortex, and to a lesser extent in the hippocampus (Fig. 8f). To estimate transduction efficiency, we quantified the percentage of GFP+ neurons in the cerebral cortex. In the four animal groups, GFP labeled approximately 10% of the total neuronal population in somatosensory cortex (SSC), where we focused our analysis due to lower transduction efficiency in visual cortex

(Fig. 8f). AD pathology in the treated mice was not significantly modified by *Kcnip4* overexpression, as no significant differences were found in amyloid plaques (determined by an anti-human amyloid beta antibody, Fig. 8g). Reactive astrogliosis, assessed by GFAP staining, remained unchanged (Fig. 8h). We observed a small but significant decrease in IBA1 staining, suggesting reduced microgliosis in *App*SAA mice overexpressing *Kcnip4* (Fig. 8i).

**Fig. 8 | AAV-mediated delivery of *Kcnip4* in excitatory neurons reduces hyperexcitability in vitro and in a humanized mouse model of AD. a** In vitro approach to evaluate AAV-mediated *Kcnip4* overexpression on neural activity in primary excitatory cortical neurons using calcium imaging. **b** Representative neuronal Ca²⁺ transients quantified as $\Delta F/F_0$ at DIV 14 for each condition. **c**, Quantification of Ca²⁺ transient frequency for each condition. Event frequency (events per minute) was averaged at the well level, with each well considered a biological replicate (4 wells per condition, 2 fields per well, 3 GFP-positive neurons per field). **d** In vivo approach to evaluate AAV-mediated *Kcnip4* overexpression in *App*^SAA and WT mice, and representative coronal section (50-μm thick) of a treated mouse illustrating transduction of cortical neurons. **e** Western blot representative image and quantification of KCNIP4 levels in cerebral cortex lysates following two different doses of *Kcnip4* AAV (*n* = 3 per group). **f** Representative images of cerebral cortex and hippocampus from *Kcnip4* AAV-treated mice and quantification of transduction efficiency of the different AAVs in SSC in WT and *App*^SAA mice. **g**

**–i** Representative images and quantification of cortical amyloid beta, GFAP, and IBA1 immunostaining in *App*^SAA mice treated with *Kcnip4* AAV or control AAV (6–7 mice per group). **j** Representative immunofluorescence image through the SSC co-stained with GFP and c-Fos. **k** Percentage of c-Fos-positive cells in all cortical neurons across study groups. **l–o** Quantification of c-Fos in GFP⁺ compared to GFP⁻ neurons from *App*^SAA and WT mice treated with *Kcnip4* AAV or control AAV (5–7 mice per group). **p** Representative immunofluorescence image through the SSC co-stained with GFP and Arc. **q** Mean Arc staining intensity in all cortical neurons across groups; **r–u**, Quantification of Arc staining intensity in GFP⁺ compared to GFP⁻ neurons from *App*^SAA and WT mice treated with *Kcnip4* AAV or control AAV (5–7 mice per group). Data are shown as median ± IQR. A two-sided t-test was used for pairwise comparisons, and one-way ANOVA with two-sided Tukey's test was used for multiple comparisons (*$p$-value < 0.05; **$p$-value < 0.01; ***$p$-value < 0.001; ****$p$-value < 0.0001; exact $p$-values are available in the Source Data file). Scale bars: 200 μm (**f**); 50 μm (**j**, **p**). Source data are provided as a Source Data file.

Finally, we quantified c-Fos and Arc, two immediate-early genes widely used as markers of neuronal activation (Fig. 8j–u, Supplementary Fig. 11b, c). These markers increase in response to excessive neuronal stimulation and seizures and have been shown to be altered in AD[49,50]. When comparing all cortical neurons in *App*^SAA and WT mice, we found elevated levels of c-Fos in *App*^SAA mice, which were reversed by *Kcnip4* AAV treatment (Fig. 8j, k). Using GFP as a marker for transduced neurons, we found that Kcnip4 AAV-mediated delivery in 12-month-old *App*^SAA mice reduced the proportion of c-Fos+ neurons in the GFP+ compared to GFP- populations (Fig. 8l–o). No significant changes in c-Fos proportions were observed in *App*^SAA mice treated with control AAV or in WT mice treated with *Kcnip4* AAV. We observed similar results for Arc expression, with reduced staining intensity in GFP+ compared to GFP- neurons in *App*^SAA mice treated with *Kcnip4* AAV and a reversal in Arc expression in treated *App*^SAA mice compared to WT controls (Fig. 8p–u). We also observed a decrease in Arc expression in WT mice treated with *Kcnip4* AAV (Fig. 8q). Thus, increased *Kcnip4* expression in excitatory cortical neurons in a humanized mouse model of AD reduced c-Fos and Arc, markers of neuronal activation and hyperexcitability, suggesting a role for *Kcnip4* in promoting resilience against hyperexcitability in AD.

## Discussion

Our strategy leveraged the spatiotemporal progression of AD to explore cellular resilience. The primary visual cortex (BA17) exhibits only mild degeneration even in end-stage AD, yet it has not been a major area of study for exploring resilience factors[9–11,51]. Layer 4 neurons, considered resilient due to low tau pathology, have not been consistently characterized in previous snRNA-seq studies in AD. We specifically identified Ex5, a cluster of L4 IT granular neurons enriched in BA17, as a resilient population that remains relatively preserved in early- and late-stage AD cortices. This resilience was linked to the upregulation of genes related to synaptic function and calcium homeostasis, including *KCNIP4*, suggesting compensatory mechanisms against hyperexcitability—an early feature in AD pathogenesis observed in human and animal models[52–54].

Building on foundational studies that have created comprehensive single-nucleus transcriptomics atlases of the human AD brain[1,2,5,7,8,23,55], our study offers a more focused analysis of resilience signatures within neocortical layer 4. While previous work broadly defined vulnerability across multiple brain regions, our approach aimed to identify specific neuronal cell types and genes linked to resilience by comparing prototype vulnerable and resilient cortices. This strategy allowed us to prioritize high-confidence genes exhibiting robust and recurrent expression changes. To achieve this, we employed unsupervised Leiden clustering followed by manual annotation. This method produced distinct neuronal clusters that are reliably distinguishable by a small, consistent set of genes (fewer than 10,

and in many cases fewer than 4), ensuring consistent assessment across the profiled neocortical regions. We further validated our annotations by using reference BICNN annotations and by comparing our clusters to high-quality reference datasets from both the prefrontal and primary visual cortices[3,5,23]. Additionally, our study provides a valuable resource through high-resolution spatial mapping of our annotated neuronal cell types on the Xenium platform. This dataset complements previous work, such as that from MERFISH in a larger AD cohort[23], and can be explored on commercial, free platforms to enable detailed analysis of specific neuronal populations and gene co-expression patterns.

Despite these strengths, our datasets have limitations. Because donors contributing to each cortical region only partially overlap, there are inherent differences in age, sex, and neuropathological severity across regional subcohorts. While these variables were included as covariates in our models, the experimental design introduces potential for residual confounding. The use of multiple sequencing technologies (Drop-seq and 10x Genomics v2 and v3) introduced technical variability. Although we addressed this through rigorous quality control and statistical modeling, residual effects of technical covariates may still influence our results, including the directionality of differentially expressed genes. BA17 nuclei were predominantly generated using Drop-seq, whereas BA9 and BA7 utilized 10x v3. Differences in sensitivity and detection efficiency between these platforms may therefore contribute to apparent regional differences. Although we applied computational batch correction and included the sequencing platform as a covariate, these technical variations may still influence comparisons of cell-type composition and gene expression. Consequently, our region-specific findings should be interpreted as hypothesis-generating. Our integrated DGE analysis framework, combining linear mixed models, bootstrap resampling, and pseudobulk-based DESeq2 may introduce specific selection biases. By requiring consensus across multiple conservative methods, our pipeline likely prioritizes 'high-confidence' genes characterized by higher baseline expression, larger effect sizes, and greater stability across donor subsets. While this approach enhances specificity and minimizes false positives, it may underrepresent subtler, context-dependent, or donor-restricted transcriptomic changes. Consequently, our findings should be viewed as a robust, conservative catalog of gene signatures rather than an exhaustive list of all pathological gene expression changes. While we modeled biological covariates like age, sex, and *APOE* status in our differential gene expression analyses, they were not explicitly included in the hdWGCNA network construction. As a correlation-based method, hdWGCNA does not natively support the inclusion of covariates in the way linear models do[56]. Therefore, some residual influence from these covariates may still affect module composition or hub gene identification. Additionally, the relatively small number of donors and the use of different donor subsets across

regions may have limited our statistical power to detect subtle changes, particularly in rare cell types or in populations that exhibit gradient-like gene expression patterns rather than distinct, well-defined clusters. For instance, we observed trends but not robust changes in highly heterogeneous populations like SST-expressing interneurons and L2/3 IT excitatory neurons despite robust data in the literature indicating their vulnerability[1,4,5,23,57]. In contrast, we identified the vulnerability of Ex3 neurons, a distinct subtype of large pyramidal cells in deep layer 3 expressing *SV2C* and heavy neurofilaments, which shows robust NFT accumulation and has been previously described to degenerate in AD in immunohistochemical studies[58]. We anticipate that increased sample sizes in future studies will allow for finer-grained mapping to high-resolution neocortical taxonomies.

Our study of L4 leverages its known cytoarchitectural variability across the neocortex. L4 is highly specialized in regions receiving topographic sensory input, such as BA17, which is characterized by a relatively thin cortical ribbon but an expanded, highly myelinated L4. This layer in BA17 features a high neuronal density and distinct sublayers that contain a dominant population of granular neurons (enriched in L4c) and smaller populations of pyramidal and giant stellate cells[19,59]. In contrast, L4 in association cortices like BA9 is thinner and often appears discontinuous, blending with pyramidal neurons of layers 3 and 5. We identified three distinct molecular subtypes of L4 excitatory neurons across these neocortical regions: Ex5 (*CUX2/RORB/EYA4/LAMA3*), Ex6 (*RORB/MME*), and Ex7 (*RORB/GABRG1*). We validated these subtypes by comparing them with publicly available datasets[3,5,8,23]. We found that the Ex5 cluster-defining genes *EYA4* and *KCNH8* preferentially label granule neurons in deep layer 4c, the same area receiving VGLUT2+ terminals from the LGN. Previous snRNA-seq studies of BA17 from healthy individuals have identified specialized L4 excitatory neuron subtypes with greater granularity[3]. Our Ex5 cluster closely matches L4_IT3, a dense pan-L4 marker, and includes L4_IT2 and L4_IT5, enriched in layers 4cβ and 4cα, respectively[3]. Although it is likely that our Ex5 cluster comprises several molecular subtypes, our approach validated L4 excitatory neuronal subtypes across neocortical regions and stages of AD progression, providing a framework for identifying gene expression changes associated with resilience.

Neuronal hyperexcitability is an early and prominent feature of AD pathogenesis, manifesting in some patients with subclinical epileptiform activity[52–54]. This state can be driven by an imbalance in excitatory and inhibitory signaling, and the subsequent gene expression changes can reflect either a maladaptive response or a compensatory, neuroprotective one. For instance, snRNA-seq profiling of cortical biopsies from living subjects with early pathology revealed electrophysiological properties and molecular signatures of pathological hyperexcitability in vulnerable L2/3 pyramidal neurons prior to their loss. That study identified the upregulation of *APP, PRNP, ATP1A3, SNAP25, SYT1,* and *CDK5* as hallmarks of this maladaptive response[57]. In contrast, our study of resilient L4 IT neurons revealed a distinct gene expression signature associated with neuroprotection. We observed the upregulation of key genes including *GRIN2A, RORA, NRXN1, NLGN1, NCAM2, FGF14, NRG3, NEGR1,* and *CSMD1*. These findings suggest that resilient neurons may activate compensatory mechanisms aimed at preventing excitotoxic damage and restoring network stability. Together, these observations are consistent with an early compensatory response in relatively resilient regions such as BA17 that becomes attenuated or fails as disease burden increases, whereas similar pathological changes emerge earlier in vulnerable regions such as BA9.

Our analysis revealed an early upregulation of *KCNIP4* in resilient Ex5 L4 IT neurons; in contrast, *KCNIP4* was downregulated in vulnerable Ex2 L2/3 IT neurons during stages of cell death, with an overall decline observed in late-stage disease. *KCNIP4* is a member of the K-channel interacting proteins (KChIPs), which include KChIP1, KChIP2, KChIP3 (DREAM/calsenilin), and KChIP4 (CALP), encoded by the *KCNIP1-4* genes[45]. KCNIP4 interacts with Kv4.2 channels, which are

key regulators of neuronal excitability. KChIP4 expression influences the subcellular localization and biophysical properties of Kv4 channels. Increased binding of KChIP4 enhances the recovery from inactivation of Kv4.2, thereby exerting an inhibitory effect on neuronal excitability[60]. KCNIP4 also interacts with presenilins, potentially modulating APP processing and Aβ levels[45,61]. Notably, KCNIP4 belongs to the recoverin branch of the EF-hand superfamily, characterized by four EF-hand calcium-binding motifs. Several members of this family have demonstrated neuroprotective properties[62]. Our results support a neuroprotective role for *KCNIP4*. Through AAV-mediated over-expression of *Kcnip4* in a humanized AD mouse model (*App*^SAA), we demonstrate a reduction in the expression of activity-dependent genes Arc and c-Fos. Our in vitro calcium imaging further confirmed that *Kcnip4* overexpression attenuated neuronal hyperexcitability, even in the presence of Aβ oligomers. While the broad AAV-mediated over-expression of *Kcnip4* across excitatory neurons in the mouse cortex does not fully recapitulate the cell-type-specific regulation observed in human AD, our data show that elevating Kcnip4 levels is sufficient to impact neuronal excitability in the context of amyloid pathology. This suggests that KCNIP4's role in regulating neuronal excitability may confer neuroprotection against excitotoxicity, particularly in response to elevated intracellular calcium levels.

Hyperexcitability has also been implicated as a pathogenic mechanism in other neurodegenerative diseases, such as amyotrophic lateral sclerosis and Huntington's disease, and is associated with aging. For example, hyperexcitability in sleep circuits can lead to sleep instability and fragmentation, particularly in older adults[54,63–65]. Thus, hyperexcitability may serve as an early biomarker of neurodegeneration and a therapeutic target. Recent interventions targeting neuronal hyperexcitability in AD include the antiepileptic drug levetiracetam and emerging non-pharmacological brain stimulation techniques[66–68]. Our study identifying neurons preserved in end-stage AD and genes associated with neuronal excitability in these cells, such as *KCNIP4*, provides insights into cellular resilience in neurodegeneration and may guide the development of interventions to slow disease progression.

## Methods

This study was conducted in accordance with all applicable ethical regulations governing the use of human tissue and laboratory animals. Postmortem human brain tissue was obtained from the UCLA Department of Pathology and Easton Center, the NIH Neurobiobank (Sepulveda repository, Los Angeles, CA [IRB: PCC#: 2015-060672, VA Project #0002] and Mt. Sinai Brain Bank, New York City, NY [IRB HAR-13-059]), and Stanford's Department of Pathology and Alzheimer's Disease Research Center (IRB IRB-33727). Informed consent for brain tissue donation was obtained in accordance with protocols approved by the respective institutions. The samples used in this study were deidentified, and the study was granted a regulatory determination of Not Human Subjects Research (NHSR). All animal procedures were performed in compliance with institutional and federal guidelines for the care and use of laboratory animals. The experimental protocols were reviewed and approved by the Administrative Panel on Laboratory Animal Care (APLAC) at Stanford University (protocol ID: 33824).

### Postmortem brain tissue

AD neuropathology was evaluated by a neuropathologist using the ABC score (National Institute on Aging and Alzheimer's Association Research Framework criteria)[10]. Relevant information, such as age, sex, ethnicity, brain weight, and postmortem interval (PMI) was recorded when available. *APOE* genotyping was performed using the SNP Genotyping service from Genewiz (Azenta Life Sciences) with genomic DNA isolated from fresh-frozen brain tissue samples. No cases with imaging or gross findings consistent with large vessel territorial

infarction, hemorrhage, primary or metastatic neoplasms, or CNS infection were included. Cases with histological evidence of hypoxic-ischemic brain injury were excluded. Tissue blocks selected for snRNA-seq underwent immunohistochemical assessment, including H&E and Nissl stains to confirm tissue integrity and the absence of microinfarcts or other focal pathologies. NeuN immunohistochemistry was performed to confirm the absence of decreased NeuN immunostaining, which could bias the sorting of NeuN$^+$ neuronal nuclei by FANS. Tau and amyloid immunohistochemistry were also performed to assess the extent of pathology in the same blocks utilized for snRNA-seq.

The tissue samples were collected from three regions: the prefrontal cortex (BA9), precuneus (BA7), and primary visual cortex (BA17), encompassing all stages of disease progression. A total of 46 donors contributed to the study (42 for BA9, 15 for BA7, and 24 for BA17). The stages of disease progression were categorized into three groups: low pathology (18 donors; 6 females, 12 males), intermediate pathology (10 donors; 7 females, 3 males), and high pathology (18 donors; 12 females, 6 males). The criteria for each group were based on the presence and distribution of tau aggregates, according to the Braak staging system[9], and of amyloid pathology, including diffuse and neuritic amyloid plaques. The density of neuritic amyloid plaques was semi-quantified using the CERAD (C) staging system[69]. The low pathology group included cases with no tau or amyloid pathology, with low AD neuropathologic change (ADNC), and cases of primary age-related tauopathy (PART), a pathology associated with aging that features NFTs with similar morphology and distribution as in AD in the absence of amyloid[70]. The PART cases in this study had a Braak stage I–III. The intermediate pathology group included cases with Braak stage III–IV and diffuse plaques or sparse (C1) neuritic plaques. The high pathology group included cases with Braak stage V–VI and moderate (C2) or abundant (C3) neuritic plaques. The mean age of the donors in the low, intermediate, and high pathology groups was 70.5 ± 9.2, 81.9 ± 13.6, and 82.4 ± 10.4 years, respectively.

RNA integrity number (RIN) was measured in all the tissue blocks selected for snRNA-seq. Total RNA extraction from ~20 mg of tissue was performed using Trizol reagent followed the RNeasy Plus Mini kit (Qiagen cat # 74134) according to the manufacturer instructions. Purified RNA was quantified using the Agilent Bioanalyzer 2100 RNA Nano chips (Agilent Technologies cat # 5067-1511) according to the manufacturer instructions. There were no significant differences in the RIN (5.8 ± 0.7, 6.2 ± 0.7, and 6.2 ± 0.7, respectively) and in the PMI (15.6 ± 8.2, 12.8 ± 8.2, and 13.8 ± 9.7 hours, respectively) between the low, intermediate, and high pathology groups.

**Single nuclear isolation and neuronal nuclei enrichment**

The fresh-frozen brain tissue blocks (~3 × 2 × 0.5 cm) were stored at −80 °C. Adequate orientation of the blocks was ensured to enable full-thickness sectioning of the cortical ribbon with a proper representation of all layers. To that end, thick sections (~500 μm) were cut spanning the entire thickness of the cerebral cortex, from the leptomeninges to the underlying white matter. The cryostat was set at −12 °C to facilitate the cutting of these thick sections while preserving the remaining tissue block frozen for further experiments. Under a stereomicroscope, the tissue slices were dissected to remove the white matter and leptomeninges. For each experiment, ~100 mg of cortical gray matter was utilized. To prevent further RNA degradation, all subsequent steps were conducted on ice under RNase-free conditions. The tissue was chopped into small pieces (<1 mm$^3$) using a chilled razor blade and homogenized with a Dounce tissue grinder (Kimble cat # 885300-0007). Each tissue sample was dissociated in 2.4 mL of homogenization buffer containing 10 mM Tris, pH 8, 5 mM MgCl$_2$, 25 mM KCl, 250 mM sucrose, 1 μM DTT, 0.5x protease inhibitor (cOmplete, Roche cat # 46931590010), 0.2 U/μL RNase inhibitor, and 0.1% Triton X-100. Typically, ~30 grinder strokes with pestle B (0.020−0.056 mm clearance) were required. Microscopic examination

using a hemocytometer was conducted to assess the number of nuclei and the presence of clumps and debris. The homogenates were subsequently filtered through a 40-μm cell strainer and transferred into two 1.5-mL Eppendorf tubes.

Iodixanol gradient centrifugation was used to further clean-up the nuclei and remove myelin debris. The homogenate was first centrifuged at 1000 × $g$ for 8 min at 4 °C. The supernatant was discarded, and the pellets were gently resuspended in 450 μL of homogenization buffer. An equal volume (450 μL) of 50% v/v iodixanol medium (41.25 mM sucrose, 24.75 mM KCl, 5 mM MgCl2, 10 mM Tris [pH 8], and 50% v/v iodixanol) was added to the homogenate and gently mixed with a pipette. The mixture was then transferred to a new 2-mL Eppendorf tube containing 900 μL of 29% iodixanol medium (125 mM sucrose, 75 mM KCl, 15 mM MgCl2, 30 mM Tris [pH 8], and 29% v/v iodixanol) by slow layering on the top. The tubes were centrifuged at 13,500 × $g$ for 20 min at 4 °C, resulting in the sedimentation of nuclei. The top layer, containing abundant myelin, and the supernatant were removed and discarded carefully, avoiding contamination of the nuclei pellet. The pellets were detached by carefully pipetting with ~50 μL of immunostaining buffer (0.1 M phosphate-buffered saline [PBS; pH 7.4], 0.5% bovine serum albumin [BSA], 5 mM MgCl$_2$, 2 U/mL DNAse I, and 0.2 U/μL RNase inhibitor), transferred to clean tubes, and gently resuspended in a total volume of 200 μL of immunostaining buffer. After a 15-min incubation with immunostaining buffer at 4 °C, with gentle rocking, NeuN primary antibody was added (mouse anti-NeuN monoclonal antibody, 1:1000, Millipore Sigma, MAB377), and incubated for 40 min at 4 °C with gentle rocking. The samples were then washed by adding 500 μL of immunostaining buffer and centrifuging at 500 × $g$ for 5 min at 4 °C. Supernatant was discarded and the pellet resuspended in immunostaining buffer containing goat-anti-mouse antibody (Alexa Fluor 647, 1:500) and a nuclear stain (Hoechst 34580; 2,5 μg/ml). Aliquots of unstained, only secondary antibody-treated, and single-stained (Hoechst, NeuN) nuclei were saved for use as controls. The number and integrity of the nuclei were evaluated microscopically after each critical step and before FANS. The typical yield for ~100 mg of cerebral cortex tissue was between 1–3 × 10$^6$ nuclei.

FANS was used to collect two single nuclear suspensions per sample (NeuN$^+$ and all nuclei). Sorting was performed using a BD FACSAria II or a Sony SH800. The sheath fluid consisted of PBS with a sheath pressure of 20 psi. Sorting was performed using a 100-μm nozzle tip or microfluidic sorting chip (100-μm). For the excitation of forward scatter (FSC) and side scatter (SSC), a 488-nm laser was employed. Hoechst 34580 and Alexa Fluor 647 were excited using 405-nm and 640-nm lasers, respectively. FANS gating was performed in the following order: FSC height vs. SSC height; SSC area vs. Hoechst fluorescence (bandpass filter 450/50); and Alexa Fluor 647 (bandpass filter 665/30) vs. Hoechst fluorescence. The FSC versus SSC gates were set with permissive limits to discard the smallest and largest particles. Hoechst fluorescence was used to distinguish single nuclei from doublets, clumps, and damaged nuclei. Alexa Fluor 647 was used to distinguish neuronal (NeuN$^+$) from non-neuronal nuclei. Controls, including unstained, only secondary antibody-treated, and only single primary antibody-treated cell suspensions, were included to adjust gates thresholds and minimize false positives from non-specific staining or autofluorescence. Two populations, all nuclei (Hoechst$^+$) and neuronal nuclei (Hoechst$^+$/NeuN$^+$), were collected. The sorted nuclear suspensions were collected in 1.5-mL Eppendorf tubes containing 100−200 μL of collection buffer consisting of 0.1 M PBS, pH 7.4, and 0.1 U/μL RNase inhibitor. After collection, BSA was added to each tube for a final concentration of 1%. To prevent nuclei from adhering to the tube walls, the collection tubes were precoated with BSA. Precoating was performed by filling the tubes with 10% BSA in PBS for 5 min, followed by rinsing with PBS and drying overnight at 4 °C.

## snRNA-seq of postmortem human brain nuclei

We used either a modified Drop-seq method[71] or the standard 10x Genomics Chromium Single Cell 3' v2 or v3 assays to profile the transcriptomes of nuclei from postmortem human brain tissue. For Drop-seq, the input single nuclei were diluted to a concentration of 200 nuclei/μl. To encapsulate individual nuclei and barcoded beads (Chemgenes, cat # Makosko-2011-10), we employed a microfluidic system (FlowJEM) and adjusted the flow parameters to generate ~100 μl (~0.5 nl) droplets (nuclei loading concentration: 200 nuclei/μl; bead concentration: 165 beads/μl; flow rate: 3 ml/h). With these parameters, both the cell occupancy and the expected doublet rates were ~5%. These rates were confirmed by observing the beads and Hoechst$^+$ nuclei within the droplets by fluorescent microscopy. Standard methods proved challenging for digesting nuclear membranes from human brain nuclei, resulting in low transcript detection. To overcome this, we tested various lysis methods (sarkosyl, SDS, and Triton) at different concentrations, with or without heat. Lysis buffers containing 1% sarkosyl yielded optimal results without disrupting droplet generation. Furthermore, brief heating of the droplet-encapsulated nuclei (5 min at 72 °C) improved lysis efficiency. Reverse transcription and PCR amplification followed previously described protocols[71]. PCR reactions, each containing 4000 beads (i.e., 200 nuclei), were individually run and subsequently pooled (typically 5–15 PCR tubes, i.e., 1000–3000 nuclei) for library preparation and sequencing.

For 10x Genomics, the input single nuclei were centrifuged at 400 ×g for 5 min at 4 °C to achieve a concentration of ~350 nuclei per μL. Nuclear concentrations were determined using a hemocytometer. On average, ~12,500 nuclei were loaded to capture around 5000 nuclei per sample (with an expected capture efficiency of ~40%). cDNA amplification and library construction followed the manufacturer's instructions.

The paired-end libraries generated by Drop-seq or 10x Genomics were sequenced on either Illumina NextSeq 500 or Novaseq 6000 platforms. A total of 243 samples (184 Drop-seq and 59 10x Genomics) were sequenced in 37 sequencing batches. For each sequencing batch, the concentration of each sample was normalized to the total number of nuclei to ensure similar numbers of reads per nucleus. Nuclei were sequenced to a depth of ~75,000 reads per nucleus.

## Preprocessing, quality control, and integration of snRNA-seq data

The paired-end raw sequence reads were preprocessed using the Kallisto bustools package (kbpython:0.26.0)[72]. An alignment index was constructed based on the human reference pre-mRNA (GRCh38, Ensembl 105). Following the Lamanno workflow, we generated separate count matrices for spliced and unspliced transcripts. These matrices were then merged to obtain the total nucleus count matrix. The quantification of total transcriptome abundance was performed for each of the three matrices. Downstream analysis, including QC, integration, cell type annotations, and differential gene expression, was performed using the unspliced transcript counts.

Empty droplets were removed by comparison with ambient RNA levels using the DropletUtils package[73]. The identification of empty droplets was performed by analyzing the knee and inflection points on the cumulative transcript counts plots for each sample individually. Nuclei with an FDR < 0.05 were removed, resulting in a total of 665,407 nuclei. Further filtering was applied to exclude nuclei with fewer than 200 genes, leaving 549,074 nuclei.

To identify potential doublets, we used the DoubletFinder package version 4.2[74]. Among the 10x Genomics samples, an average doublet rate of 2.85% and 1.74% was detected in v2 and v3 samples, respectively, while the Dropseq samples had a doublet rate of 0.003%. The identified doublets were labeled and retained during batch correction and data integration. Following clustering and dataset annotation, the majority of labeled doublets clustered together. These clusters, containing doublets, were excluded from further downstream analysis.

To analyze the raw count data, we used Scanpy in the Python package version 3.9.1[75]. First, we used a series of preprocessing steps for normalization and scaling. Highly variable genes were identified using default parameters and a dispersion threshold of 0.5. Principal Component Analysis (PCA) was applied to reduce dimensionality, generating 50 principal components. Subsequently, a neighborhood graph was constructed using default parameters with 15 neighbors, and Leiden graph-based clustering was performed using correlation distance metrics. To address batch effects and integrate data from different brain regions and disease stages, we used Harmony (v1.2.2769)[76]. Integration was based on silhouette score values of 0.8 or higher, as well as visual inspection of UMAP plots representing experimental assay, sequencing batch, donor, brain region, disease stage, sex, and UMI abundance. The selected integration variables included the experimental assay (Dropseq, 10x Genomics v2, and 10x Genomics v3) and brain region (BA9, BA7, and BA17). After integration, the neighborhood graph and Leiden graph-based clustering were generated again. Marker genes for each cluster were determined using the Wilcoxon rank sum test with a significance threshold of adjusted p-value (padj) <0.05.

The integrated dataset contained 549,074 nuclei. To further optimize the lower gene cutoff, thresholds from 200 to 500 genes were tested, and 300 genes were chosen as the final cutoff. All clusters comprised nuclei from every donor, and no clusters exclusively contained nuclei with low UMI counts. Three small clusters containing doublets, totaling 17,442 nuclei, were excluded. Mitochondrial gene content was measured and annotated, but only outlier nuclei with higher than 5% mitochondrial genes (1778 nuclei) were discarded, resulting in a total of 424,528 high-quality nuclei (362,224 neuronal and 62,304 non-neuronal) for downstream analysis.

## Major cell type, neuronal subtype, and glial cell state annotations

The major neuronal and non-neuronal populations were identified based on the expression of known marker genes: *SLC17A7* (excitatory neuron), *GAD1* (inhibitory neuron), *FGFR3*, *AQP4*, and GFAP (astrocyte), *CSF1R*, *CX3CR1*, and *CD163* (microglia), *PLP1* and *MOG* (oligodendrocyte), *PDGFRA* and *CSPG4* (OPC), *CLDN5* and *FLT1* (endothelial), *NOTCH3* (pericyte), and *CYP1B1* and *COL15A1* (VLMC).

These major cell type clusters were subsetted and reclustered within the integrated PC space to identify neuronal subtypes and glial states. Clustering reliability was determined based on silhouette score values of 0.8 or higher and WCSS (Within Cluster Sum of Squares). The first 30 PCs and a resolution of 1.0 were employed for both the excitatory subset (282,930 nuclei) and the inhibitory subset (79,294 nuclei). Marker genes for each cluster were ranked using the Wilcoxon rank sum test with the following criteria: minimum expression fraction (either in the tested cluster or in all other nuclei combined) of 0.2, log-fold change > 0.5, padj <0.05. After merging two excitatory clusters that lacked marker genes to reliably distinguish between them and discarding two small inhibitory clusters (207 nuclei) with mixed markers, a total of 18 excitatory (Ex) and 19 inhibitory (In) clusters were obtained. We visualized the UMAP with a minimum distance of 0.6 and a spread of 1.4.

To identify the top marker genes for each cluster, the following criteria were applied: expression fraction within the cluster (pts) > 0.2; expression fraction within all other nuclei (pts_rest) <0.1; ratio pts/pts_rest > 3; log-fold change >1.5; and padj <0.05. For Ex1, Ex2, and Ex5, the pts_rest was set at <0.2, and the ratio pts/pts_rest was set at > 2. The clusters were named based on canonical markers for major subclasses (i.e., *CUX2*, *RORB*, *THEMIS*, and *FEZF2* for excitatory; and *LHX6* and *ADARB2* for inhibitory), followed by 1–3 top marker genes. Additionally, we compiled gene sets consisting of 7–10 genes for each neuronal

subtype, selected from the top marker genes. These marker genes and gene sets precisely labeled each of our annotated neuronal subtypes in our dataset and a reference dataset[23], and thus are useful to identify neuronal subtypes computationally and by histology.

To compare our neuronal clusters and their marker genes with a reference dataset, we utilized a publicly available dataset containing over one million neuronal nuclei from the DLPFC of donors with dementia and healthy controls[23]. To determine the degree of similarity between the annotations in the two datasets, we subset both count matrices keeping only highly variable genes (3000 genes), identified the top 10 markers for each cluster, and calculated the cosine similarity distance between the mean expression values of genes for each cluster. A lower distance in the similarity matrix indicates a higher level of agreement.

The non-neuronal subset clusters included: astrocytes (14,691), microglia (5071), oligodendrocytes (36,589), OPCs (5770), and vascular cells (183). These populations were reclustered using the first 10 PCs, with a resolution of 0.3 for astrocytes, 0.2 for microglia and oligodendrocytes, and 0.1 for other types. The top marker genes for each cluster were identified using the same method as for the neuronal clusters, using the following thresholds: pts > 0.2; pts_rest <0.1; ratio of pts/pts_rest > 2.5; log-fold change > 1.0; and padj <0.05. We annotated 4 astrocyte, 4 microglia, and 2 oligodendrocyte cell states. Other non-neuronal types were not subclassified further due to their relatively low nucleus numbers.

### Cell identity prediction using scANVI

We employed the scANVI[77] (version 1.0) machine learning method to predict the identity of unannotated neuronal nuclei with relatively low UMI counts from our dataset and to predict the identity of neuronal populations from public datasets using the annotations from our dataset. For model training, we utilized our excitatory (18 neuronal clusters) and inhibitory (19 neuronal clusters) datasets as training set. We selected 2000 highly variable genes and the top 200 marker genes for each cluster (Wilcoxon rank sum test, log-fold change > 0.8, padj <0.05), along with our cell-type-specific gene sets. The model underwent training for a maximum 200 epochs with 3 layers and 50 latent spaces. To address batch effects during training, we introduced a combined batch effect key considering both the profiling assay (DropSeq, 10x v2, 10x v3) and brain region (BA9, BA7, and BA17). We monitored model convergence and loss for each epoch using an elbow plot to determine the optimal number of epochs for effective convergence. To prevent overfitting, we employed a single-layer perceptron with the 'linear_classifier' parameter set to 'True', promoting model simplicity and reducing bias towards the training data. Additionally, we applied the var activation (torch.nn.functional.softplus) function to ensure stable optimization. To represent rare cell types adequately, we set the 'n_samples_per_label' to 1000. Probabilities of cell cluster assignments from the latent space were computed using the 'soft' function, providing a confidence measure for each prediction. Model accuracy was assessed by comparing true labels with predictions using sensitivity and specificity measures. Additionally, we conducted differential testing using the Wilcoxon rank sum test, employing predicted annotations to validate the reliability of the neuronal subtypes inferred from our predictions. We assigned identity to query data using a probability threshold of 0.99 to minimize false positives. We predicted excitatory cell identity labels for five public reference datasets (SEA-AD DLPFC and MTG; Mathys et al., 2023; Green et al., 2024; Jorstad et al. 2023)[3,5,8,23] to demonstrate that the pretrained model can be applied across datasets.

### Neuronal subtype proportion quantification

To quantify the relative proportions of each neuronal subtype within each disease group (low, intermediate, high pathology) and region (BA9, BA17), we calculated the relative abundance of each subtype per

donor in relation to the total number of neurons. Analyses were conducted separately for the BA9 and BA17 brain regions at the layer-annotated neuronal cluster level, restricted to cells expressing a minimum of 500 genes per nucleus. To test for significant differences in cell composition among disease groups, we conducted two complementary analyses: scCODA[34], a Bayesian modeling framework tailored for compositional single-cell data, and a generalized linear mixed model (GLMM) for modeling variances and covariances.

For scCODA (version 0.1.9), the model formula used was:

Cell type counts ~ Disease group + Sex + APOE genotype + Assay + Age

The software was run in automatic reference mode, allowing it to select a stable reference cell type that is assumed not to vary across conditions. Donor age was standardized and included as a numeric covariate. We utilized this Bayesian framework to assess changes in relative cell type abundances, focusing on posterior inclusion probabilities and credible intervals to determine statistically significant differences. To prepare the data for analysis, raw count matrices were constructed for each brain region, representing the number of cells per donor and neuronal subtype combination based on layer-specific annotations. Following this, cell type proportions were computed, and the Bayesian model was fitted to assess the effects of disease progression while adjusting for sex, APOE genotype, assay platform, and age.

Additionally, to evaluate the robustness of our compositional inferences, we performed a stress test sensitivity analysis using a modified version of scCODA restricted to neuronal populations only. Neuronal nuclei were filtered using a quality-control threshold of n_genes > 500 prior to aggregation. Donor-level cell type counts were modeled using scCODA's Dirichlet–multinomial framework. To take into account compositional effects and the sparsity-inducing prior, we reparameterized the regression coefficients ($\beta$) using a HalfNormal prior with an explicit sign constraint, such that all coefficients were restricted to be non-positive ($\beta \leq 0$). Under this formulation, inferred effects represent neuronal losses only relative to the reference population, and apparent expansions of individual neuron types are not permitted. The loss-only model was fitted using the same covariate structure as the primary analysis. Aside from smaller HMC step sizes required for stable sampling under the constrained parameterization, all modeling choices were held constant. This analysis therefore, represents a deliberately conservative stress test rather than an alternative generative model.

We applied GLMMs to evaluate differences in neuronal subtype representation across disease groups, using proportional abundance as the outcome variable. The outcome variable was the fraction of each neuronal subtype per donor, bounded between 0 and 1, and modeled using a beta distribution with a logit link. Each neuronal subtype was modeled independently. The main effect tested was the disease group, with the low pathology group as the reference. Additional fixed effects in the models included assay platform (Dropseq, 10x Genomics v2, and 10x Genomics v3), age, sex, and APOE genotype. Donor was included as a random intercept to account for repeated measures and inter-individual variation. Neuronal subtypes represented in fewer than three unique donors were excluded to ensure robust model fitting. All modeling was conducted using the glmmTMB package[78] (version 1.1.11) in R, with data preprocessing performed using dplyr and tidyr. For each model, we extracted fixed effect estimates, standard errors, z-statistics, and associated p-values. Results were compiled across neuronal subtypes and exported for interpretation.

### Spatial transcriptomics using Visium

We used the 10x Genomics Visium platform (Spatial 3' v1 chemistry) to spatially map 37 neuronal subtypes (18 excitatory and 19 inhibitory), astrocytes, oligodendrocytes, OPCs, and microglia in fresh-frozen tissue sections from the neocortex of AD and healthy control donors. A

total of 16 tissue sections were studied, including controls (BA9, BA7, BA17, motor cortex, entorhinal cortex, and hippocampus from a control donor, and two additional sections of BA9 and BA7 from another donor) as well as AD (BA9 and BA7 from 3 donors with high AD pathology and BA9 from 2 donors with intermediate pathology). The sections were cut at a thickness of 12 μm on a cryostat and mounted on fiduciary frames of four 10x slides. H&E staining was performed, and the slides were temporarily coverslipped with mounting medium (85% glycerol containing 0.2 U/μL RNAse inhibitor) and digitally scanned at 200x magnification using a Zeiss Axio Imager M2 microscope equipped with a color digital camera (Axiocam) and MBF Stereo Investigator with a 2D slide scanning extension module. Permeabilization enzyme treatment was applied to the tissue for 15 minutes at 37 °C, as determined by the Tissue Optimization protocol provided by 10x Visium. Reverse transcription, second strand synthesis, and cDNA amplification were carried out according to the manufacturer's recommendations. We utilized the targeted Human Neuroscience gene expression panel, which consists of 1186 genes, and supplemented it with a custom panel comprising 197 cell type-specific marker genes. The marker genes were selected from our snRNA-seq dataset based on their specificity to label our annotated neuronal clusters and their expression levels. The custom hybridization capture panel oligos were obtained from IDT (IDT NGS Discovery Pools). Library sequencing was performed on the Illumina Novaseq 6000 platform at a depth of 15,000 reads per spot, resulting in a sequencing saturation of approximately 96%. This 1383 gene panel provides a cost-efficient tool for mapping neuronal vulnerability in the human AD brain while allowing to sequence at a 90% lower cost compared to whole transcriptome sequencing. To evaluate the quality of our spatial data, we used the 10x Space Ranger pipeline, which maps the transcriptomic data on the high-resolution microscopic images. On average, we detected 1336 out of the total 1383 targeted genes. The median number of targeted genes and UMI counts detected per spot were 221 and 392, respectively.

## Integration of snRNA-seq and Visium spatial transcriptomics

We used Stereoscope[79] to integrate the spatial transcriptomics (ST) data generated with 10x Genomics Visium and the snRNA-seq data. First, we subsetted the snRNA-seq data based on the genes present in the Visium spatial transcriptomics dataset (1383 genes). We trained variation auto encoder model using the snRNA-seq data to construct a single-cell reference latent variable for inferring cell type-specific gene expression patterns. For model training, we used the following parameters: layer = 'unspliced'; labels_key = 'Author_Annotation'; max epochs = 200. We checked the model convergence by elbow plot. Then, we trained the spatial model using the Visium data and the pre-trained snRNA-seq data for a maximum of 2000 epochs. This allowed us to identify cell types using negative binomial latent variables. We performed different iterations to visualize cell populations in the Visium space considering different combinations of cell types (i.e., all major cell types; each of the excitatory neurons, inhibitory neurons, and glial populations; and each excitatory neuronal subtype). To visualize each of the cell subtypes in each ST tissue section slide, we utilized the matplotlib and seaborn plotting Python packages.

## Spatial transcriptomics using Xenium

We used the 10x Genomics Xenium spatial platform to map our annotated neuronal cell subtypes at single-cell resolution. A total of 16 human brain sections were analyzed, 8 from BA9 and 8 from BA17, including samples from 4 donors with high AD pathology and 4 control donors. Fresh-frozen brain sections were cut at a thickness of 10 μm on a cryostat and mounted inside the fiducial frames of four Xenium slides. Tissue section fixation and permeabilization were performed according to the manufacturer's protocol. We used the predesigned 266-gene Xenium Human Brain Gene Expression panel along with a custom 100-gene panel. To ensure consistency across samples, probe

hybridization, ligation, and rolling circle amplification were performed for all four slides together using a HybEZ™ II Oven (ACD Bio). Additionally, the Cell Segmentation Add-on Kit was employed for multimodal segmentation, following the manufacturer's protocol for staining. Background fluorescence was chemically quenched according to manufacturer's instructions. Imaging, signal detection, and spatial decoding were performed using the Xenium Analyzer (10x Genomics) under standard settings.

## Xenium data preprocessing and neuronal cell subtype annotation

Spatial transcriptomic data generated with the Xenium platform were preprocessed using the Xenium Ranger (version 3.1) with squidpy[80] (version 1.2.3) standard pipelines. Cell segmentation was performed using the multimodal cell segmentation algorithm, with the final segmentation prioritized using first the interior RNA staining (ribosomal RNA) to delineate cellular boundaries, followed by an isotropic nuclear (DAPI) expansion of 5 μm. The latter primarily identified small cells, particularly glia, that exhibited low ribosomal RNA staining.

We first annotated major cell types using a strategy aimed at addressing transcript signal overlap between cells in close proximity. We implemented four independent approaches: (1) manual annotation based on k-nearest neighbor graphs, Leiden clustering, and canonical marker genes; (2) heuristic classification using a custom Python script to assign cell types based on the highest-expressed transcripts; (3) deep neural network (DNN) classification via spatialID[81] (version 1.0.0), trained on the SEA-AD DLPFC dataset[23]; and (4) ingest-based label transfer directly projecting SEA-AD DLPFC annotations onto the spatial data. Predictions from these methods were integrated using an ensemble voting strategy, generating consensus annotations and confidence scores. Cells with a consensus confidence greater than 0.5 were retained for downstream analyses.

Next, we performed neuronal cell subtype annotation using ingest-based label transfer with our snRNA-seq dataset as a reference. Neurons with more than 50 transcripts were annotated. Shared genes between Xenium and snRNA-seq were identified and subsetted from both datasets. PCA was performed on the reference dataset (adata_nucleus) using the top 15 principal components to construct a k-nearest neighbors graph (k = 20). The Xenium data (adata_xenium) were then projected into the reference PCA space, and cell labels were transferred using sc.tl.ingest, based on mutual nearest neighbors in the PCA embedding. The labels were transferred separately for each tissue. This approach enabled efficient and accurate label transfer while preserving fine neuronal subtype resolution.

## Co-expression network analysis

We performed weighted gene co-expression network analysis (WGCNA) on our high-dimensional snRNA-seq data using the hdWGCNA R package[56] (version 0.2.23) to compute co-expression gene modules of interconnected genes within each neuronal cell subtype and brain region. We constructed meta-cells using the following parameters: group.by = "Author_Annotation", k = 25, and minimum cell threshold of 50. After constructing the meta-cells, we normalized the object using default parameters, including the use of Harmony for dimensional reduction and batch correction. We then constructed co-expression networks. Gene correlations were transformed into a similarity matrix using the power function, which preserves strong correlations. Modules were identified through hierarchical clustering with similarity distance measures. Module reliability and robustness were estimated using bootstrap resampling with 5000 iterations. We ranked the highly correlated genes, defined as the kME (module eigengene), by their kME values for each neuronal cell subtype within the modules and retained the top 50 intra-module co-expressed genes.

## Differential gene expression analysis

We performed differential gene expression (DGE) analysis to compare the low vs intermediate pathology groups (designated as 'early' changes) and the intermediate vs high pathology groups (designated as 'late' changes), within each neuronal subtype and brain region. To ensure the reliability of our analysis, we employed various methods, including a zero-inflated regression mixed model implemented in MAST[47] (version 1.24.1) and lme4[82] (version 1.1-34), bootstrap resampling with 100 iterations, and pyDESeq2[83] (version 0.3.5) on pseudo-bulk aggregated counts.

For the zero-inflated regression mixed model (MAST and lme4), we used the following model formula:

Zlm(~condition + (1 | donor) + cngeneson + Assay + Age + Sex + RIN + total_counts, sca, method = 'glmer')

In this model, donor is considered a random effect. The fixed covariates include "cngeneson" (i.e., cellular gene detection rate), age, sex, RIN, and total raw sequencing counts. The DE genes were filtered using the following thresholds: percentage of expression > 20% for at least one condition, |logFC|>0.1, and false discovery rate <0.05.

To ensure the robust and reproducible identification of significant DE genes, we employed bootstrapping followed by DGE (MAST/lme4, as detailed above). This resampling technique mitigates potential effects from outliers and the varying number of nuclei per cluster. We conducted a series of 100 iterations. In each iteration, we randomly selected 50% of nuclei from each neuronal cell subtype from each comparison. Subsequently, we computed the DE genes for each iteration and assigned confidence scores based on frequency analysis across all iterations. We filtered the DE genes based on their consistent identification as differentially expressed, retaining those genes that exhibited the same significant up or downregulation in at least 20 out of the 100 iterations.

Additionally, we used a pseudobulk aggregation method with raw gene abundance counts to construct representative expression profiles for each neuronal subtype within each donor, disease group, and brain region. The data were organized with donors as rows and genes as columns. Next, we aggregated the data from individual donors into a single pseudobulk count dataset. We then performed log normalization on the raw count data and applied a gene filter, retaining only genes expressed in at least 20 nuclei. Subsequently, we created a DESeq2 object using pyDESeq2. To evaluate the data's inherent variability, we conducted PCA, a high-dimensional reduction technique. For DGE analysis, we established a design matrix to compare disease groups. To ensure statistical significance, we applied the Benjamin-Hochberg correction method with a threshold of padj <0.05.

We defined 'high-confidence' DE genes as those identified by at least two different methods: the mixed model and either one of the other DGE methods (bootstrap or pseudobulk) or network co-expression analysis (top 50 genes by kME values from the hdWGCNA). This comprehensive approach aimed to enhance the reliability of our DGE predictions by ensuring consistency across methods and experimental conditions.

## Functional enrichment analysis

We used multiple methods for functional enrichment analysis, including Enrichr[84] (GSEApy, version 1.0.6), Metascape[85], and g:profiler[86]. The input data consisted of high-confidence DE genes obtained from comparing low vs. intermediate and intermediate vs. high pathology groups ('early' and 'late' changes) within each neuronal subtype and brain region. For Enrichr[84], we used brain-specific gene sets from the Genotype-Tissue Expression (GTEx)[87] and Synaptic Gene Ontology (SynGO)[88] databases to establish background gene expression. Statistical significance thresholds were determined using an adjusted p-value < 0.05 and at least a minimum of three genes per group. We utilized Metascape with the following custom parameters: a minimum overlap of 5, a p-value threshold of 0.01, and a minimum

enrichment score of 2.5. The top 50 enriched functional modules were visualized in heatmaps using the Matplotlib and Seaborn Python packages.

Additionally, we used g:profiler to perform functional enrichment analysis for Ex2 L2/3 IT and Ex5 L4 IT, using as input the top 50 co-expressed network genes from each module from our hdWGCNA analysis. We selected key driver GO terms within the Molecular Function and Biological Process categories, with GO terms having a size between 10 and 1000 and an adjusted p-value threshold of 0.05. The top enrichment terms from each module were visualized in a dot plot using Matplotlib and Seaborn, and the enrichment networks were visualized using Cytoscape[89].

## Immunohistochemistry in human brain tissue

Immunofluorescence to quantify KCNIP4 was performed on 20 μm-thick cryosections of fresh-frozen brain tissue. Sections were fixed with 2% PFA for 20 minutes, blocked with 10% normal goat serum (NGS) and 2% BSA in PBS with 0.25% Triton X-100 (PBT) for 1 hour at RT, and then incubated overnight at 4 °C with primary antibodies in PBT containing 3% NGS and 0.5% BSA. The primary antibodies used were guinea pig anti-NeuN (1:50, Sigma ABN90), rabbit anti-EYA4 (1:50, Thermo Fisher Scientific PA552113), and mouse anti-KCNIP4 (1:50, Proteintech 60133-1-Ig). After three washes with PBT, sections were incubated with secondary antibodies for 2 hours at RT: Alexa Fluor 647 anti-guinea pig (Invitrogen, A21450; 1:200), Alexa Fluor 488 anti-rabbit (1:200, Invitrogen A11070;), and Alexa Fluor 546 anti-mouse (Invitrogen, A11018; 1:200). Sections were counterstained with DAPI (1:2000, Invitrogen) for 20 minutes, rinsed in PBS, mounted with aqueous mounting medium (Invitrogen P36930), and sealed. Images were acquired using a Zeiss LSM980 laser scanning confocal microscope with consistent parameters and processed with CellProfiler using custom pipelines for automatic cell segmentation based on NeuN and analysis of EYA4-positive cells and KCNIP4 intensity (available in the GitHub repository). We quantified one section per donor, with an average of approximately 250 excitatory neurons per case. Neurons were identified by morphology and marker expression, and quantification was restricted to well-defined regions of interest within the cortical layers.

Immunohistochemistry for VGLUT2 and NeuN was performed on 50 μm-thick free-floating fixed sections, obtained from tissue blocks fixed with 4% PFA for 3 days, cryoprotected in 30% sucrose, and sectioned on a sliding microtome. The free-floating sections were rinsed in PBT and incubated in PBT containing 1% hydrogen peroxide for 30 minutes to block endogenous peroxidase activity. After washing with PBT, sections were incubated in a blocking solution containing 10% NGS and 2% BSA for 1 hour at RT. Sections were then incubated with mouse anti-NeuN (1:1000, Millipore Sigma MAB377) or mouse anti-VGLUT2 (1:1000, Millipore Sigma MAB5504) diluted in PBT containing 3% NGS and 0.5% BSA overnight at 4 °C. After washing, sections were incubated with a biotinylated goat anti-mouse antibody (VectorLabs BA-9200) diluted in PBT containing 3% NGS for 2 hours at RT, and then washed again. This was followed by incubation with ABC solution (Vectastain Elite ABC-HRP kit, VectorLabs PK-6100) for 1 hour at RT. For the chromogenic reaction, 3,3'-Diaminobenzidine (DAB) substrate solution (Sigma D5905) was used. Sections were air-dried, dehydrated with ethanol followed by xylene, and coverslipped with Permount mounting medium.

## RNAscope ISH in human brain tissue

Double fluorescent RNAscope ISH staining was performed on 20-μm-thick cryosections from fresh-frozen tissue, following the manufacturer's protocol (Multiplex Fluorescent Reagent Kit v2 #323100). Human RNAscope probes were obtained from ACD Bio. to detect the following genes: *EYA4* (#510931), *MME* (#410891), *GABRG1* (#485931), and *SLC17A7* (#415611). Opal reagents from Akoya Biosciences were used for fluorescence detection: Opal 690 (FP1497001KT; for *EYA4*),

Opal 570 (FP1488001KT; for *MME* and *GABRG1*) and Opal 520 (FP1487001KT; for *SLC17A7*). DAPI signal served as the anatomical reference to identify cortical layers. Images were acquired using a Zeiss LSM980 laser scanning confocal microscope with consistent parameters. Experiments were conducted in duplicate using 2 sections from each of 2 healthy control donors for both BA9 and BA17.

## Experimental animals

*App* knock-in mice (B6.Cg-*App*[tm1.1Dnli]/J, strain #034711; also known as *App*[SAA]) and WT controls (C57BL/6J, strain #000664) were obtained from The Jackson Laboratory (Bar Harbor, ME, USA). The mice were bred on a pure BL6/J background, and genotypes were confirmed by real time PCR (Transnetyx, Cordova, TN). All mice were housed in a barrier facility with ad libitum access to standard chow and water, on a 12:12-h light:dark cycle, and euthanized at study endpoints by trans-cardiac perfusion under deep anesthesia, according to the guidelines for animal testing and research under a protocol approved by the Stanford's Administrative Panel on Laboratory Animal Care (APLAC). Previous studies have reported sex differences in some behavioral assays, but not in pathology[24]. Due to the small sample size, we used only male mice in this study.

## Primary mouse cortical neuron culture

Primary cortical neurons were isolated from postnatal day 0 (P0) C57BL/6 mouse pups following established protocols[90]. Briefly, cortices from 5–8 pups were dissected and enzymatically dissociated using trypsin and DNase I, followed by mechanical trituration with fire-polished glass pipettes. The resulting cell suspension was plated at a density of 80,000 cells/cm$^2$ in glass bottom 24-well plates (Cellvis P24-1.5H-N) coated with poly-L-lysine and maintained in serum-free Neurobasal medium (Gibco) supplemented with GlutaMAX (Gibco), B27 (Gibco), and penicillin/streptomycin. Cultures were incubated at 37 °C in a humidified atmosphere of 5% CO$_2$, with half of the medium replaced every 3 days.

## Calcium imaging in primary mouse cortical neurons

Neurons were transduced on day in vitro 7 (DIV7) with AAV-PHP.eB-CaMKIIa-*Kcnip4*-P2A-EGFP or control AAV-PHP.eB-CaMKIIa-EGFP (Addgene #50469-PHPeB) at a multiplicity of infection (MOI) of 5000 viral genomes (vg)/cell. Cloning and AAV production were performed by Vector Biolabs (mouse *Kcnip4* isoform 1; NCBI Reference Sequence: NP_001186171.1). To enable calcium imaging, cells were co-transduced with AAV-PHP.eB-Syn.NES-jRGECO1a.WPRE.SV40 (Addgene #100854-PHPeB) at the same MOI. After 12 hours of incubation with AAVs, media was fully replaced with maintenance media mixed 1:1 with pre-conditioned media (collected during previous media changes and stored at −20 °C) to minimize AAV-associated toxicity while preserving growth factors. On DIV12, intracellular calcium levels were increased by treating the cells with 200 nM amyloid-β (1–42) oligomers (Anaspec) or with PBS as a control. Oligomers were prepared by dissolving 0.5 mg amyloid-β (1–42) in 1% NH$_4$OH to yield a 1 mM stock, followed by a 1:10 dilution in PBS and incubation at 37 °C for 24 h in a thermocycler.

Time-lapse imaging of jRGECO1a fluorescence was performed at DIV14 at 5 Hz for 100 seconds per field using a Zeiss LSM980 in wide-field fluorescence mode under controlled environmental conditions (37 °C, 5% CO$_2$). Regions of interest (ROIs) were manually drawn around neuronal somas using the Multi-Measure tool in ImageJ, and mean intensity values were extracted per frame. Traces were obtained from three GFP-positive neurons per field (two fields per well, four wells per condition). Fluorescence changes ($\Delta F/F_0$) were calculated using a rolling baseline defined as the 10th percentile over a 10-second window. Calcium transients were detected as events exceeding a threshold of 0.2 $\Delta F/F_0$, selected based on visual inspection and applied uniformly across conditions. Event frequency (events per minute) was averaged at the well level, with each well considered a biological replicate. Data analysis, including trace extraction and event detection, was performed using Python 3.11.12.

## TUNEL assay in primary mouse cortical neurons

Immediately following calcium imaging, neurons were washed three times with PBS and fixed with 4% PFA/4% Sucrose in PBS for 20 min at RT. After three additional washes, cells were stored in PBS at 4 °C for no longer than 24 h. TUNEL staining was performed using the Click-iT™ Plus TUNEL assay (Thermo Fisher Scientific, C10247 Far-Red). Briefly, cells were permeabilized with 0.25% Triton™ X-100 in PBS for 20 min at RT, incubated with the TdT reaction cocktail for 1 h at 37 °C, and subsequently with the Click-iT™ reaction cocktail for 30 min at RT protected from light. After TUNEL labeling, cells were washed in PBS and blocked in 10% NGS in PBS for 30 min at RT, followed by incubation with anti-GFP antibody (1:1000; Invitrogen, A11122) in 5% NGS for 1 h at RT. After three PBS washes (5 min each), cells were incubated with Alexa Fluor 488-conjugated anti-rabbit secondary antibody (1:1000; Invitrogen, A11070) and DAPI (1:5,000) in 5% NGS for 1 h at RT. Cells were then washed three times in PBS and stored at 4 °C (up to 48 h) prior to imaging. Images were acquired using a Zeiss LSM980 laser scanning confocal microscope with identical acquisition settings across conditions. TUNEL-positive, GFP-expressing neurons were manually counted using ImageJ (three fields per well, four wells per condition, each well was considered a biological replicate).

## AAV-driven KCNIP4 expression in excitatory neurons in adult mice

Twelve-month-old *App*[SAA] and control male mice were injected retro-orbitally with $1 \times 10^{11}$ vg of AAV-PHP.eB-CaMKIIa-*Kcnip4*-P2A-EGFP in 100 μL of PBS or with AAV-PHP.eB-CaMKIIa-EGFP as a control. Thirty days post-injection, mice were perfused with 0.9% saline for 3 minutes, and the brains were extracted. The right hemisphere was frozen in isopentane at −50 °C, while the left hemisphere was fixed in 4% PFA for 24 hours, followed by cryoprotection in 30% sucrose for 24 hours. The fixed tissue was cut into 50 μm-thick free-floating sections and stored at −20 °C in a cryoprotectant solution (30% glycerol, 30% ethylene glycol in PBS) until further processing.

## Immunoblotting of mouse cortex tissue

Frozen mouse brain cortex was lysed in tris/SDS/glycerol buffer, and protein concentration was determined using a BCA assay (Thermo Fisher Scientific). Twenty μg of protein were separated on 10% Mini-PROTEAN® TGX™ Precast Protein Gels (Bio-Rad) and transferred to PVDF membranes using the Trans-Blot Turbo (Bio-Rad) semi-dry transfer system. The membranes were blocked with 5% milk in tris-buffered saline with 0.05% Tween 20 (TBS-T) for 60 minutes at RT and then incubated overnight at 4 °C with mouse anti-Kcnip4 antibody (1:1000, Proteintech 60133-1-Ig) and anti-GAPDH antibody (1:10,000, Thermo Fisher Scientific MA5-15738). After three 10-minute washes with TBS-T, the membranes were incubated with a goat anti-mouse secondary antibody (1:1000, Invitrogen G-21040) for 60 minutes at RT. Following washing with TBS-T, membranes were developed using ECL (Thermo Fisher Scientific, 32106) and imaged on X-ray film (Thermo Fisher Scientific, 34091). Images were processed and quantified using ImageJ.

## Immunohistochemistry in mouse brain tissue

Immunohistochemistry was performed on 50-μm-thick free-floating slices of mouse brain tissue. For immunofluorescence, slices stored in cryoprotectant solution were washed three times for 10 minutes each with PBS, then photobleached under full spectrum LED light for 48 hours in a cold chamber[4]. The sections were blocked with 10% NGS and 2% BSA in PBT for 1 hour at RT and incubated overnight at 4 °C with primary antibodies in PBT containing 3% NGS and 0.5% BSA. For c-Fos and Arc quantification, we used: guinea pig anti-c-Fos (1:200, Synaptic

Systems 226 308), mouse anti-Arc (1:200, Synaptic Systems 156 111), rabbit anti-GFP (1:1000, Invitrogen A11122), and anti-NeuN (1:200, Millipore Sigma ABN90P). After primary antibody incubation, sections were washed three times for 15 minutes each with PBT and then incubated with secondary antibodies (1:200, Alexa Fluor 647 anti-guinea pig, Invitrogen A21450; Alexa Fluor 488 anti-rabbit, Invitrogen A11070) for 2 hours at RT. Following three additional 15-minute washes, tissues were counterstained with DAPI (1:2000, Invitrogen) for 30 minutes at RT. The slices were rinsed in 0.05 M TBS, mounted with aqueous mounting medium (P36930, Invitrogen), and sealed. Images were acquired using a Zeiss LSM980 laser scanning confocal microscope with consistent parameters across all samples. We analyzed SSC due to better transduction efficiency in this region. Image processing was carried out using CellProfiler with custom pipelines for automatic segmentation of GFP⁺ and GFP⁻ neurons based on NeuN and GFP markers, and quantification of c-Fos and Arc (available in the GitHub repository).

For chromogenic immunohistochemistry, free-floating sections were incubated in 0.6% hydrogen peroxide in PBT for 20 minutes to block endogenous peroxidase activity. The primary antibodies used included rabbit anti-human amyloid beta (1:500, IBL 18584), rabbit anti-GFAP (1:2000, Dako Z0334), and rabbit anti-Iba1 (1:500, FujiFilm 019-19741). Sections were then incubated with a secondary antibody, followed by an avidin/biotin-based peroxidase system and chromogenic detection using DAB, as previously described for human tissue. Brightfield images were captured with a Zeiss Axio Imager 2 and a Hamamatsu digital camera (C11440), and the stained cortical area was quantified using ImageJ with automated thresholding.

## Statistics & Reproducibility
Statistical analyses were performed using Prism 10 (GraphPad Software) unless otherwise stated in specific Methods sections. Sample sizes were chosen based on previous publications in the field. Biological replicates were analyzed to assess the biological variability and reproducibility of data; the distinctions between technical and biological replicates are explained in each section of the methods. Experimental mice from all genotypes or conditions were processed together; mice were randomly assigned to experimental groups, using littermates for different groups whenever feasible. Investigators were not blinded to experimental groups during data analysis. Samples were tested for normality using the Shapiro-Wilk normality test. Outliers were screened using the ROUT method (Q = 1%), and no data points were excluded from mouse and in vitro experiments. Unless otherwise stated, data were analyzed by t-test or ANOVA followed by post hoc Tukey's test to compare multiple samples. Differences were considered significant when $p$-values < 0.05. Statistical details of experiments are described in the figure legends.

## Reporting summary
Further information on research design is available in the Nature Portfolio Reporting Summary linked to this article.

## Data availability
The raw snRNA-seq data, associated metadata, and processed digital expression matrices have been deposited at the NCBI's Gene Expression Omnibus with accession number GSE263468. Eight of 243 samples were included in previous studies (GSE129308 and GSE181715). The snRNA-seq datasets are publicly available for interactive viewing and exploration on the Cellxgene platform at https://cellxgene.cziscience.com/collections/0d35c0fd-ef0b-4b70-bce6-645a4660e5fa. The Xenium dataset is publicly available at Zenodo: https://zenodo.org/records/16703438. Source data are provided with this paper.

## Code availability
The scripts and the pretrained models are available at GitHub and accessible at Zenodo: https://doi.org/10.5281/zenodo.18113528[91]

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

## Acknowledgements

Human tissue was obtained from Stanford's Department of Pathology and Alzheimer's Disease Research Center (NIH/NIA P30AG066515), UCLA Department of Pathology and Easton Center, and the NIH Neurobiobank (Sepulveda repository in Los Angeles, CA and Mt. Sinai Brain Bank in New York City, NY). This work was supported by grants to I.C. from NIH/NIA (R01AG059848, R01AG082147), BrightFocus (A20173465), the Alzheimer's Association (AARG-17-528298), and the Chan Zuckerberg Initiative (Ben Barres Early Career Acceleration Award, grant ID 199150).

## Author contributions

Conceptualization: S.A.P.D., I.C.; Human tissue procurement and Neuropathology: K.V., I.C.; Single-nuclear transcriptomics data generation: M.O.G., J.P.; Spatial transcriptomics data generation: J.P., J.S.R; Data analysis: S.A.P.D., W.T.; Histology: J.S.R., J.P., K.V., Y.C.L.; Functional assays in mice: J.S.R.; Funding acquisition: I.C.; Supervision: I.C.; Writing: S.A.P.D., J.S.R., I.C.; All authors read and approved the final manuscript.

## Competing interests

The authors declare no competing interests.
