## [Transparent Peer Review file · Nature Communications]

Molecular Signatures of Resilience to Alzheimer's Disease in Neocortical Layer 4 Neurons

Corresponding Author: Dr Inma Cobos

Version 0:

Reviewer comments:

Reviewer #1

(Remarks to the Author)

No additional comments and revisions requested. I commend the authors on addressing all of our prior comments and look forward to seeing this work published.

Reviewer #2

(Remarks to the Author)

Reviewer #3

(Remarks to the Author)

The authors' findings addressed a critical question in the field of Alzheimer's disease (AD), i.e., why neocortical layer 4 neurons are relatively resilient to AD. Furthermore, they identified the important role of Kcnp4 contributing to this resilience. In addition, the authors have successfully addressed most of the critical comments from previous reviewers, especially with new experimental data and new computation analysis. These findings will complement recent multiomics studies focusing on selective neuronal vulnerability in AD.

The revision supports the conclusions and claims.

The data analysis looks good to me.

The methodology used in this study sound appropriate and meet the expected standards in the field of AD.

There are enough details provided in the methods for the work to be reproduced.

Reviewer #4

(Remarks to the Author)

Dharshini et al have made substantial improvements in the revised manuscript, including better characterizing their cell type taxonomy with spatial transcriptomics, improving their methods descriptions, and clarifying several points of confusion in the text. However, several comments raised previously were not adequately addressed. Chief among them is the lack of new snRNAseq data generation in BA17 (the primary focus of the manuscript), which is mostly profiled with an older sequencing technology and in a subset of donors with higher pathology. There remain substantial confounds within the data that can only be addressed with prominent caveats and which raise questions on its broader utility. That leaves the finding that KCNIP4 may be an important regulator of neuronal hyperexcitability in AD as the primary draw of the manuscript. The authors have taken steps to improve these experiments and demonstrate this mechanism with an orthogonal in vitro approach. The mechanism is not entirely surprising though, as prior work established that KCNIP4 regulates Kv4 channels

(albeit not in an in vivo context).

1. Experimental design & data quality

- a. The authors have only partially addressed my comment. Including covariates in a linear model may not account for differences in the underlying distribution of those covariates across brain regions due to differing donor subsets. For example, based on the new Supplementary Figure 1e, there appear to be more nuclei from higher pathology donors in BA7 and BA17 than in BA9. The authors address some of these caveats in their added discussion paragraph (Page 14, Lines 14-18), though should also acknowledge that spurious associations could arise from their limited and non-overlapping donor subsets. These are critical caveats in the experimental design that, at a minimum, should also be acknowledged clearly at the beginning of the manuscript and potentially inline with the results of both the differential compositional and expression analyses. This concern could also be mitigated if the authors observe the same associations in the donors that do overlap across regions, though they may not have enough data to achieve statistical significance.
- b. If the authors add the caveat above at the beginning of the manuscript this comment will be adequately addressed.
- c. The thresholds provided in Supplementary Table 2 are helpful. A threshold of 500 genes detected/nuclei is still very low though, particularly for excitatory neurons profiled with 10xv3. At that threshold, there is a large difference in the number of nuclei retained in 10xv3 (85%) and Dropseq (42%). This is particularly concerning because most nuclei sampled in BA17 are from Dropseq, while most nuclei from BA9 are from 10xv3. You can also see a poor integration between modalities in Supplementary Figure 1d for excitatory neurons. As above, this creates a confound and could all have large effects on the results, which should be clearly stated at the start of the manuscript and potentially inline with the results of both the differential compositional and expression analyses.

2. Cellular Annotation

- a. This comment is adequately addressed, the authors should be commended for trying to relate their labels as best as possible to current taxonomies. It is worth noting though that needing to use less fine-grained labels speaks to the limited quantity of data available.
- b. This comment is adequately addressed.
- c. While the authors state defining V1C types is not the primary goal of their manuscript, the use of their own cell type taxonomy necessitates carefully characterizing that taxonomy. The authors should be commended for including new Xenium data to do so. However, their response has not adequately addressed my comment. While Jorstad et al define V1 specialized cell types as those where >60% of the cells come from V1, the L4 IT populations are much closer to being entirely absent or present in visual cortex. This is consistent with the impression left by the authors' new Xenium data presented in Figure 3f, where there are almost no Ex5 nuclei in BA9. The authors point out that Ex5 is less abundant in BA9 vs BA17 (~30% versus ~4%), though it is still one of the more abundant types in BA9 in their snRNAseq taxonomy in Figure 4a. This suggests an inconsistency in the cell type annotation that may be lumping transcriptionally distinct populations together inappropriately. The Xenium data raises similar concerns about other populations that may be inappropriately merged as well. For example, Ex2 is in layer 3 in BA9, but layer 3 and 4 in BA17. This was also not observed in Jorstad et al.

3. Cell type compositional analysis

- a. The caveat addresses this comment, though many of the problems noted above would be greatly mitigated by additional 10v3 data from BA17.
- b. This comment is adequately addressed.
- c. scCODA models the counts of each cell type, not relative abundances as the latter are susceptible to compositional effects. The model is likely returning a strong positive association for a single neuron group rather than a weak negative association from the multiple types because of its strong prior that fewer types are more likely to drive change. To address this properly, the model needs to be modified such that neurons can only be lost (e.g. the prior on the beta coefficient is set to HalfNormal instead of Normal). This is also why using a GLMM on the proportions is inappropriate without some stabilizing transformation like a centered log ratio (as implemented in tools like crumblr).
- d. This comment is adequately addressed.

4. Differential gene expression

- a. A line emphasizing that changes in "late" BA17 likely relate to "early" BA9 after introducing the groups in the results section would greatly improve readers' understanding why you focused on certain groups of genes. If added, this comment will be adequately addressed.
- b. A more clear statement of early compensatory response and then its failure later in the discussion section would greatly improve readers' understanding and address this comment. As is, there is some reading between the lines that must be done.
- c. This comment has not been adequately addressed. By grouping genes with different dynamics together you obfuscate complex dynamics within major biological pathways. If, for example, genes within "synaptic vesicle cycle" have opposing dynamics it is unclear exactly what is occurring, even if the majority are downregulated (such that the module on average has a negative logFC). Rather than finding the "overarching themes", how do the authors know they are not simply enriching for pathways with genes that tend to be expressed by neurons?
- d. The Supplementary Table provided by the authors is helpful. It suggests the authors may consider simplifying their procedure to remove bootstrapping as it does not appear to be recovering as many genes as pseudobulk or hdWGCNA (though not necessary). As originally requested though, the authors should add discussion to the manuscript about how a complex gene selection process could bias the genes that are selected. Alternatively, they can run their downstream analyses on gene sets from each methodology to show they are robust as suggested in their reply.
- e. This comment is adequately addressed.
- f. This comment has not been adequately addressed. As originally requested (and given the large discrepancy in sequencing depth across technologies) the authors should include an extended data or supplementary figure showing how coefficients on UMI counts and sequencing chemistry are weighted in their models.
- g. The above figure should include a comparison for how these coefficients are weighted in Ex5 versus other types to account for the concern originally raised.

h. As above, these caveats should be made inline in the results section.

i. This comment is adequately addressed.

j. This comment is adequately addressed.

5. Increased KCNIP4 expression and mouse modeling

a. This comment is adequately addressed. The authors efforts are admirable and it is unfortunate existing datasets were not of sufficient quality to support replication analyses.

b. The authors should include these details of their quantification in the methods section of the manuscript, otherwise this comment is adequately addressed. It is unfortunate that KCNIP4 probes failed, though the effort is appreciated.

c. The authors should include these details of their quantification in the methods section of the manuscript and explicitly state in the results (and methods) their experiments were performed in SSC due to technical limitations of transduction in mouse visual cortex.

d. This comment is adequately addressed.

e. This comment is adequately addressed.

All other minor comments were adequately addressed. One more small minor comment: there appears to be a title duplication on Page 6 Line 10 and Page 7 Line 28.

Molecular Signatures of Resilience to Alzheimer's Disease in Neocortical Layer 4 Neurons

NCOMMS-25-87580-T

We thank the reviewers for their positive feedback and for endorsing our work. We appreciate the time and effort they spent reviewing our manuscript and the constructive comments they provided during the revision process, which helped improve the quality of the paper. Below, we provide a point-by-point response to the additional suggestions and questions raised by Reviewer #4. Changes in the manuscript are highlighted in blue.

REVIEWERS' COMMENTS

Reviewer #1

No additional comments and revisions requested. I commend the authors on addressing all of our prior comments and look forward to seeing this work published.

Reviewer #3

The authors' findings addressed a critical question in the field of Alzheimer's disease (AD), i.e., why neocortical layer 4 neurons are relatively resilient to AD. Furthermore, they identified the important role of Kcnp4 contributing to this resilience. In addition, the authors have successfully addressed most of the critical comments from previous reviewers, especially with new experimental data and new computation analysis. These findings will complement recent multiomics studies focusing on selective neuronal vulnerability in AD.

The revision supports the conclusions and claims.

The data analysis looks good to me.

The methodology used in this study sound appropriate and meet the expected standards in the field of AD.

There are enough details provided in the methods for the work to be reproduced.

Reviewer #4

Dharshini et al have made substantial improvements in the revised manuscript, including better characterizing their cell type taxonomy with spatial transcriptomics, improving their methods descriptions, and clarifying several points of confusion in the text. However, several comments raised previously were not adequately addressed. Chief among them is the lack of new snRNAseq data generation in BA17 (the primary focus of the manuscript), which is mostly profiled with an older sequencing technology and in a subset of donors with higher pathology. There remain substantial confounds within the data that can only be addressed with prominent caveats and which raise questions on its broader utility. That leaves the finding that KCNIP4 may be an important regulator of neuronal hyperexcitability in AD as the primary draw of the manuscript. The authors have taken steps to improve these experiments and demonstrate this mechanism with an orthogonal in vitro approach. The mechanism is not entirely surprising though, as prior work established that KCNIP4 regulates Kv4 channels (albeit not in an in vivo context).

Response: We thank the reviewer for the thorough re-evaluation of our manuscript and for acknowledging the substantial improvements made regarding the spatial transcriptomics characterization and method descriptions.

We also appreciate and acknowledge the reviewer's remaining concerns and the limitations of our study related to sequencing technology and the distribution of donor pathology. We have followed the reviewer's advice to explicitly state these limitations in the manuscript. We believe these transparency measures will ensure that readers interpret the data in the appropriate context.

We are pleased that the reviewer recognizes the value of our findings regarding KCNIP4 as a regulator of neuronal hyperexcitability. We have made every effort to address the remaining textual and analytical points raised below.

1. Experimental design & data quality

a. The authors have only partially addressed my comment. Including covariates in a linear model may not account for differences in the underlying distribution of those covariates across brain regions due to differing donor subsets. For example, based on the new Supplementary Figure 1e, there appear to be more nuclei from higher pathology donors in BA7 and BA17 than in BA9. The authors address some of these caveats in their added discussion paragraph (Page 14, Lines 14-18), though should also acknowledge that spurious associations could arise from their limited and non-overlapping donor subsets. These are critical caveats in the experimental design that, at a minimum, should also be acknowledged clearly at the beginning of the manuscript and potentially inline with the results of both the differential compositional and expression analyses. This concern could also be mitigated if the authors observe the same associations in the donors that do overlap across regions, though they may not have enough data to achieve statistical significance.

b. If the authors add the caveat above at the beginning of the manuscript this comment will be adequately addressed.

c. The thresholds provided in Supplementary Table 2 are helpful. A threshold of 500 genes detected/nuclei is still very low though, particularly for excitatory neurons profiled with 10xv3. At that threshold, there is a large difference in the number of nuclei retained in 10xv3 (85%) and Dropseq (42%). This is particularly concerning because most nuclei sampled in BA17 are from Dropseq, while most nuclei from BA9 are from 10xv3. You can also see a poor integration between modalities in Supplementary Figure 1d for excitatory neurons. As above, this creates a confound and could all have large effects on the results, which should be clearly stated at the start of the manuscript and potentially inline with the results of both the differential compositional and expression analyses.

Response: We acknowledge the reviewer's point that linear covariates may not fully account for differences in the distributions of age, sex, and pathology across region-specific donor subsets. Because the donor cohorts are non-overlapping across regions, residual confounding could persist. We have revised the manuscript to explicitly acknowledge this limitation in Results and Discussion (**Pages 4, 8 and 14**). A statement has been added indicating that the observed region-pathology associations may be influenced by region-specific sampling and should be interpreted as hypothesis-generating. As noted by the reviewer, there is insufficient data from donors contributing to all regions to achieve statistical significance.

We also agree with the reviewer that the correlation between brain region and sequencing platform (Drop-seq for BA17 vs. 10x v3 for BA9) may represent a technical confound that biases region-level comparisons. As suggested, we have acknowledged this limitation in Results and Discussion (**Pages 8 and 14**).

2. Cellular Annotation

- a. This comment is adequately addressed, the authors should be commended for trying to relate their labels as best as possible to current taxonomies. It is worth noting though that needing to use less fine-grained labels speaks to the limited quantity of data available.
- b. This comment is adequately addressed.
- c. While the authors state defining V1C types is not the primary goal of their manuscript, the use of their own cell type taxonomy necessitates carefully characterizing that taxonomy. The authors should be commended for including new Xenium data to do so. However, their response has not adequately addressed my comment. While Jorstad et al define V1 specialized cell types as those where >60% of the cells come from V1, the L4 IT populations are much closer to being entirely absent or present in visual cortex. This is consistent with the impression left by the authors' new Xenium data presented in Figure 3f, where there are almost no Ex5 nuclei in BA9. The authors point out that Ex5 is less abundant in BA9 vs BA17 (~30% versus ~4%), though it is still one of the more abundant types in BA9 in their snRNAseq taxonomy in Figure 4a. This suggests an inconsistency in the cell type annotation that may be lumping transcriptionally distinct populations together inappropriately. The Xenium data raises similar concerns about other populations that may be inappropriately merged as well. For example, Ex2 is in layer 3 in BA9, but layer 3 and 4 in BA17. This was also not observed in Jorstad et al.

Response: Xenium data in Fig. 3f show that Ex5 is abundant in BA17 (V1) but rare in BA9, and present only at low levels in BA18 (V2), consistent with strong enrichment in the visual cortex. Although Ex5-like nuclei are detected in BA9 by our snRNA-seq data, their low spatial abundance in Xenium highlights a distinction between molecular identity and regional prevalence. These Ex5-like cells share a substantial molecular concordance with Ex5 in BA17, including canonical markers (*RORB*, *CUX2*, *CUX1*, *EYA4*, *LAMA3*, *TRPC3*). We agree that Ex5 is best described as a V1-specialized cell population, with rare but transcriptionally similar counterparts outside V1. We have clarified this point in the revised manuscript (**Page 7**).

3. Cell type compositional analysis

- a. The caveat addresses this comment, though many of the problems noted above would be greatly mitigated by additional 10v3 data from BA17.
- b. This comment is adequately addressed.
- c. scCODA models the counts of each cell type, not relative abundances as the latter are susceptible to compositional effects. The model is likely returning a strong positive association for a single neuron group rather than a weak negative association from the multiple types because of its strong prior that fewer types are more likely to drive change. To address this properly, the model needs to be modified such that neurons can only be lost (e.g. the prior on the beta coefficient is set to HalfNormal instead of Normal). This is also why using a GLMM on the proportions is inappropriate without some stabilizing transformation like a centered log ratio (as implemented in tools like *crumblR*).
- d. This comment is adequately addressed.

Response: We thank the reviewer for the insightful comments regarding the use of scCODA and its sparsity-inducing prior. We agree that such priors can, in some settings, favor explanations involving apparent expansion of a single population rather than coordinated reductions across multiple cell types. To directly address this concern, we performed a conservative sensitivity analysis restricted to neuronal populations. Specifically, we modified the scCODA framework to impose a “loss-only” constraint by reparameterizing the regression coefficients (β) with a HalfNormal prior and enforcing $\beta \leq 0$, thereby

allowing only disease-associated decreases in abundance and explicitly disallowing expansion as an explanatory mechanism.

Under this restrictive formulation, our primary biological conclusions remained robust. In both BA9 and BA17, multiple excitatory populations, including L2/3 IT and L5 IT neurons, showed statistically credible decreases. Among inhibitory neurons, SST-expressing interneurons exhibited selective vulnerability in BA9, consistent with prior reports. In contrast, L4 IT neurons did not reach statistical significance for loss under this constraint, indicating that their behavior in the unconstrained model is likely influenced by compositional context rather than reflecting a statistically supported decrease.

Together, this sensitivity analysis supports the interpretation that the observed vulnerabilities in L2/3 IT, L5 IT, and SST interneurons represent disease associated losses, while clarifying the interpretation of L4 IT neurons in compositional models. The full results of this analysis are provided in **Supplementary Table 5**.

4. Differential gene expression

a. A line emphasizing that changes in “late” BA17 likely relate to “early” BA9 after introducing the groups in the results section would greatly improve readers’ understanding why you focused on certain groups of genes. If added, this comment will be adequately addressed.

Response: We agree with the reviewer about this probable association and have included a sentence in the results section after introducing the groups for DGE: “Given AD progression, we expect that gene expression changes observed in late-stage BA17 will be concordant with those seen in early-stage BA9” (**Page 9**).

b. A more clear statement of early compensatory response and then its failure later in the discussion section would greatly improve readers’ understanding and address this comment. As is, there is some reading between the lines that must be done.

Response: We thank the reviewer for this suggestion. We have revised the Discussion to clearly articulate a staged regional pattern in which BA9 exhibits disease-associated changes earlier, whereas BA17 shows evidence of an initial compensatory response that is lost at later stages (**Page 16**). Specifically, in BA17 we observe early enrichment of pathways related to synaptic function, calcium signaling, and calcium homeostasis, followed at higher pathology by downregulation of these same pathways. This pattern is consistent with the interpretation that initially protective or homeostatic mechanisms become insufficient as disease burden increases.

c. This comment has not been adequately addressed. By grouping genes with different dynamics together you obfuscate complex dynamics within major biological pathways. If, for example, genes within “synaptic vesicle cycle” have opposing dynamics it is unclear exactly what is occurring, even if the majority are downregulated (such that the module on average has a negative logFC). Rather than finding the “overarching themes”, how do the authors know they are not simply enriching for pathways with genes that tend to be expressed by neurons?

Response: We acknowledge that grouping genes with differing dynamics can obscure within-pathway heterogeneity and have clarified this point in the legend for figure 5 (**Page 45**). For Fig. 6a,b, enrichment was performed separately for each comparison using only high-confidence DE genes identified within that specific region, stage, and neuronal subtype, rather than all neuronal genes. For each enriched term, Metascape provides the DE genes contributing to each enriched term, and for visualization we compute a signed pathway score by averaging the log₂ fold changes of the term’s contributing DE genes within each comparison. A negative score indicates net downregulation within

the term; a positive score indicates net upregulation. Thus, the module value reflects a net directional bias among DEGs, not uniform regulation of all genes in the pathway. For example, a term such as synaptic vesicle cycle may include both up- and downregulated genes; a negative score means that downregulated genes predominate, even if some genes are upregulated. Importantly, enrichment is not driven by generic neuronal expression, since terms arise from DE genes within specific neuronal subtypes and show distinct directionality across comparisons (e.g., BA9 vs BA17, early vs late), which would not be expected if enrichment were due to nonspecific neuron-expressed genes alone.

d. The Supplementary Table provided by the authors is helpful. It suggests the authors may consider simplifying their procedure to remove bootstrapping as it does not appear to be recovering as many genes as pseudobulk or hdWGCNA (though not necessary). As originally requested though, the authors should add discussion to the manuscript about how a complex gene selection process could bias the genes that are selected. Alternatively, they can run their downstream analyses on gene sets from each methodology to show they are robust as suggested in their reply.

Response: While we acknowledge that bootstrapping yields fewer genes than pseudobulk or hdWGCNA, we chose to retain this approach as a rigorous method that prioritizes stability and specificity over breadth of discovery. As suggested, we have added a comment to the Discussion (**Page.14**) regarding how our multi-step selection process might bias the final gene list toward high-confidence genes (those with higher baseline expression, larger effect sizes, and donor consistency) at the potential cost of underrepresenting subtler or donor-specific changes.

e. This comment is adequately addressed.

f. This comment has not been adequately addressed. As originally requested (and given the large discrepancy in sequencing depth across technologies) the authors should include an extended data or supplementary figure showing how coefficients on UMI counts and sequencing chemistry are weighted in their models.

g. The above figure should include a comparison for how these coefficients are weighted in Ex5 versus other types to account for the concern originally raised.

Response: We thank the reviewer for this suggestion. To address the potential impact of large differences in sequencing depth across technologies, we now include an additional panel in **Supplementary Figure 9b** that summarizes the fitted coefficients for sequencing depth (log-transformed UMI counts) from the linear mixed models. This panel shows the per-gene distributions of the corresponding nuisance covariate coefficients, providing a visualization of how sequencing depth are weighted in the model. As suggested, this panel explicitly compares these coefficients between Ex5 neurons and other excitatory neuron types. Across regions and disease stages, the distributions of coefficients for log(UMI) are highly similar between Ex5 and other excitatory neurons, with comparable medians and dispersion.

h. As above, these caveats should be made inline in the results section.

i. This comment is adequately addressed.

j. This comment is adequately addressed.

5. Increased KCNIP4 expression and mouse modeling

a. This comment is adequately addressed. The authors efforts are admirable and it is unfortunate existing datasets were not of sufficient quality to support replication analyses.

b. The authors should include these details of their quantification in the methods section of the manuscript, otherwise this comment is adequately addressed. It is unfortunate that KCNIP4 probes

failed, though the effort is appreciated.

c. The authors should include these details of their quantification in the methods section of the manuscript and explicitly state in the results (and methods) their experiments were performed in SSC due to technical limitations of transduction in mouse visual cortex.

d. This comment is adequately addressed.

e. This comment is adequately addressed.

Response: We thank the reviewer for their understanding of the challenges with replication in independent datasets as well as with probe optimization, and for their positive evaluation of our experimental efforts. Regarding the specific requests in points 5b and 5c, we have updated the manuscript to address these points with textual revisions. We have included detailed quantification procedures in the Methods section. Additionally, we explicitly state in both the Results and Methods sections that the *in vivo* experiments were performed in the somatosensory cortex (SSC), rather than the visual cortex, due to the technical limitations of viral transduction in the mouse visual cortex (**Page 12**). Of note, the SSC and visual cortex are both primary sensory cortices with a prominent layer 4 that receives afferents from primary thalamic sensory nuclei (VPM/VPL for somatosensory, LGN for visual).

All other minor comments were adequately addressed. One more small minor comment: there appears to be a title duplication on Page 6 Line 10 and Page 7 Line 28.

Response: Thank you for noting this. It has been corrected in the revised version.